# Sex dependence of opioid-mediated responses to subanesthetic ketamine in rats

Tommaso Di Ianni [1,6] ✉, Sedona N. Ewbank [1], Marjorie R. Levinstein [2], Matine M. Azadian [1], Reece C. Budinich[2], Michael Michaelides [2,3] & Raag D. Airan [1,4,5] ✉

Subanesthetic ketamine is increasingly used for the treatment of varied psychiatric conditions, both on- and off-label. While it is commonly classified as an *N*-methyl D-aspartate receptor (NMDAR) antagonist, our picture of ketamine's mechanistic underpinnings is incomplete. Recent clinical evidence has indicated, controversially, that a component of the efficacy of subanesthetic ketamine may be opioid dependent. Using pharmacological functional ultrasound imaging in rats, we found that blocking opioid receptors suppressed neurophysiologic changes evoked by ketamine, but not by a more selective NMDAR antagonist, in limbic regions implicated in the pathophysiology of depression and in reward processing. Importantly, this opioid-dependent response was strongly sex-dependent, as it was not evident in female subjects and was fully reversed by surgical removal of the male gonads. We observed similar sex-dependent effects of opioid blockade affecting ketamine-evoked postsynaptic density and behavioral sensitization, as well as in opioid blockade-induced changes in opioid receptor density. Together, these results underscore the potential for ketamine to induce its affective responses via opioid signaling, and indicate that this opioid dependence may be strongly influenced by subject sex. These factors should be more directly assessed in future clinical trials.

A sub-anesthetic dose of (*R,S*)-ketamine (or ketamine) rapidly and robustly attenuates depressive symptoms[1,2], yielding much recent enthusiasm for the treatment of varied neuropsychiatric disorders and leading to the FDA approval of the (*S*)-ketamine stereoisomer for treatment-resistant depression[3]. Notwithstanding, our understanding of the mechanism of action of subanesthetic ketamine has remained elusive, calling for further investigation to better elucidate the mechanistic underpinnings of its therapeutic effects.

The therapeutic action of ketamine is commonly attributed to its non-competitive antagonism at the glutamatergic *N*-methyl D-aspartate receptors (NMDAR)[4,5], but our picture of its underlying mechanisms is incomplete[6,7]. Other experimental drugs acting via selective NMDAR antagonism or other downstream glutamate-mediated effects have provided minimal efficacy in clinical trials in psychiatry[8], while NMDAR-independent pathways have also been suggested to mediate the antidepressant action of ketamine[9]. Recent

[1]Department of Radiology, Stanford University School of Medicine, Stanford, CA 94305, USA. [2]Biobehavioral Imaging and Molecular Neuropsychopharmacology Unit, National Institute on Drug Abuse Intramural Research Program, Baltimore, MD 21224, USA. [3]Department of Psychiatry and Behavioral Sciences, Johns Hopkins University School of Medicine, Baltimore, MD 21205, USA. [4]Department of Materials Science and Engineering, Stanford University School of Medicine, Stanford, CA 94305, USA. [5]Department of Psychiatry and Behavioral Sciences, Stanford University School of Medicine, Stanford, CA 94305, USA. [6]Present address: Departments of Psychiatry & Behavioral Sciences and Radiology & Biomedical Imaging, University of California, San Francisco, San Francisco, CA 94143, USA. ✉e-mail: tommaso.diianni@ucsf.edu; rairan@stanford.edu

clinical evidence suggests that pretreatment with naltrexone, a non-selective opioid receptor antagonist, attenuates the antidepressant effect of intravenous (i.v.) ketamine in humans[10,11]. However, subsequent studies challenged these findings[12,13], warranting further investigation. Preclinical data also show diverging evidence: opioid receptor blockade suppressed the behavioral and neurophysiological responses to ketamine and its enantiomers in some studies[14–17], while others reported no significant effects[18]. The potentially pivotal role of the opioid system in the antidepressant efficacy of ketamine raises concern for ketamine's abuse liability and potential for dependence[17,19], as opioid signaling is thought to mediate the hedonic aspects of reward processing[20] and the reinforcing effects of drugs of abuse[21,22]. This concern is particularly relevant in light of the ongoing opioid crisis[23]. In addition, there has been relatively limited assessment of other biological variables, including subject sex, that may explain the heterogeneity of clinical and preclinical findings of the potential opioid dependence of ketamine's therapeutic efficacy and adverse effects.

Here we sought to determine how opioid receptor blockade affects ketamine-evoked neural activity changes. We imaged neural responses to ketamine in awake-restrained male and female rats using functional ultrasound imaging (fUSI). This neuroimaging modality is based on neurovascular coupling and closely tracks neural activity by way of high-resolution whole-brain maps of cerebral blood volume (CBV)[24–26]. Notably, our awake restraint imaging paradigm provides an opportunity to assess ketamine action in the context of an acute stress model, as transient neurophysiological changes and cognitive behavioral adaptations have been shown to take place during and immediately after acute exposure to restraint stress in rodents[27–29]. Finally, we establish that ketamine's acute opioid-mediated effects on neural activity are reflected in physiologic changes at the synaptic level and in the expression of behavioral sensitization, and we investigate the molecular mechanisms causing the observed opioid-dependent behavioral adaptations.

## Results

### Functional ultrasound imaging of intravenous subanesthetic ketamine administration

We first determined that fUSI could resolve the acute effects of subanesthetic ketamine administration. We prepared male rats with a surgical craniotomy to allow ultrasound penetration and implanted a chronic polymeric prosthesis to enable repeated imaging (Fig. 1a). With the animals awake and restrained, we continuously recorded fUSI images at 2.5 mm rostral and 3.5 mm caudal to bregma (Fig. 1b). To avoid excessively large craniotomies, the two brain slices were imaged in different animals, and this factor was taken into account in the subsequent statistical analyses, as appropriate. Ketamine (10 mg/kg, i.v.) evoked a rapid and sustained increase (peak at 3–5 min; >50 min duration) in CBV signal that extended over cortical and subcortical regions (Fig. 1c and Supplementary Movie 1). We then infused varied doses of ketamine (0, 1, 5, and 10 mg/kg, i.v.) to determine the dose-response relationship of the CBV changes. We segmented the CBV signals in regions of interest (ROIs) obtained by registering the relevant slices from the Paxinos & Watson rat brain atlas[30] onto a power Doppler vascular template (Supplementary Fig. 1) and calculated the mean regional CBV time series (Fig. 1d, Supplementary Fig. 2). The time series presented a clear dose-response relationship, confirmed by statistical comparisons of the peak CBV (Fig. 1e) and area under the curve (AUC; Supplementary Fig. 2b). Importantly, neither the peak CBV nor AUC were significantly correlated to the local vascularization level, as measured by the regional baseline power Doppler signal, suggesting that the recorded changes were independent of the intrinsic vascular anatomy (Supplementary Fig. 2c). These results demonstrate that fUSI is able to image the neural effects evoked by acute ketamine infusions

with a dose-dependent response, and confirm previous findings using different imaging modalities[16,31].

### Pharmaco-fUSI closely tracks ketamine-evoked gamma-band power in the prefrontal cortex

To further confirm the validity of our pharmaco-fUSI modality, we performed electrocorticography (ECoG) measurements over the cingulate area 1 (Cg1) of the medial prefrontal cortex (mPFC) (Fig. 2a). Ketamine administration (10 mg/kg, i.v.) acutely increased electrophysiological responses in the delta/theta (1-8 HZ), alpha (8–12 Hz), beta (12–30 Hz), and gamma (30–80 Hz) bands (Fig. 2b, c). Power changes dissipated rapidly (within 10 min post ketamine) in the delta/theta, alpha, and beta bands. As expected[9,32], gamma-band power showed a sustained response that lasted for the entire duration of the session (50 min).

We then regressed the normalized ECoG power changes in each frequency band and the Cg1 CBV signal using a four-parameter gamma-distribution function (Fig. 2d, Supplementary Fig. 3c, d). The curve fitting was performed via a least-squares minimization routine, and the regressed β values were statistically compared between the ECoG spectral bands and the Cg1 CBV signal. There were significant differences between Cg1 CBV and delta/theta (corrected $P < 7.91E-07$), alpha ($P < 6.26E-07$), and beta ($P < 0.0176$) bands (Fig. 2d), whereas no statistical significance was observed when comparing Cg1 CBV to gamma-band power ($P > 0.0756$). Comparisons of the least-squares residuals were in all cases not significant (Supplementary Fig. 3f; $P > 0.11$), indicating that the differences in the β values were caused by actual differences in the regressed time series rather than by variability in the goodness of fit. We also performed ECoG recordings and least-squares regressions with 1 mg/kg i.v. ketamine administration (Supplementary Fig. 3a, b, e). Although the regression did not produce reliable results for the delta/theta, alpha, and beta bands, as indicated by the high residuals (Supplementary Fig. 3f), this lower ketamine dose also showed a high degree of correlation between the gamma-band power and the Cg1 CBV time series (Supplementary Fig. 3a, b). Our observations are in agreement with a previous study reporting on the high correlation of fUSI signals with gamma (30-90 Hz) and high gamma-band (110-170 Hz) local field potentials[33]. In addition, our results are further reinforced by prior evidence using functional MRI in humans and non-human primates[34–36]. Altogether, these results confirm that the acute responses to subanesthetic ketamine recorded by our pharmaco-fUSI modality are driven by neural activity changes and not necessarily by a non-specific cardiovascular effect.

### Opioid receptor blockade modulates ketamine responses in male, but not in female, rats

Following these initial methodologic and dose-response characterizations of using fUSI to study the effects of subanesthetic ketamine, we selected 10 mg/kg as the dose of ketamine for subsequent experiments, following prior rodent studies of the affective effects of ketamine that show reliable behavioral efficacy with this dose[9]. To map the presence of region-specific opioid-mediated effects, we pretreated two groups of male and female rats with subcutaneous (s.c.) injections of either vehicle (VEH; saline) or naltrexone (NTX; 10 mg/kg) followed by i.v. ketamine (KET; 10 mg/kg) or saline after 10 min (Fig. 3a). The 10 mg/kg naltrexone dose yields near complete mu opioid receptor occupancy in the mouse brain[37] and blocked the effect of (S)-ketamine on acute locomotion[16]. Each rat was imaged three times under the treatment conditions of VEH + KET, NTX + KET, and NTX + VEH in a three-arm crossover design. Treatment conditions were assigned in randomized order to control for possible effects of prior drug exposure, and we allowed for a 7-day washout period between ketamine injections for full drug clearance.

Functional maps contrasting the NTX + KET and VEH + KET groups revealed region-specific effects of naltrexone pretreatment

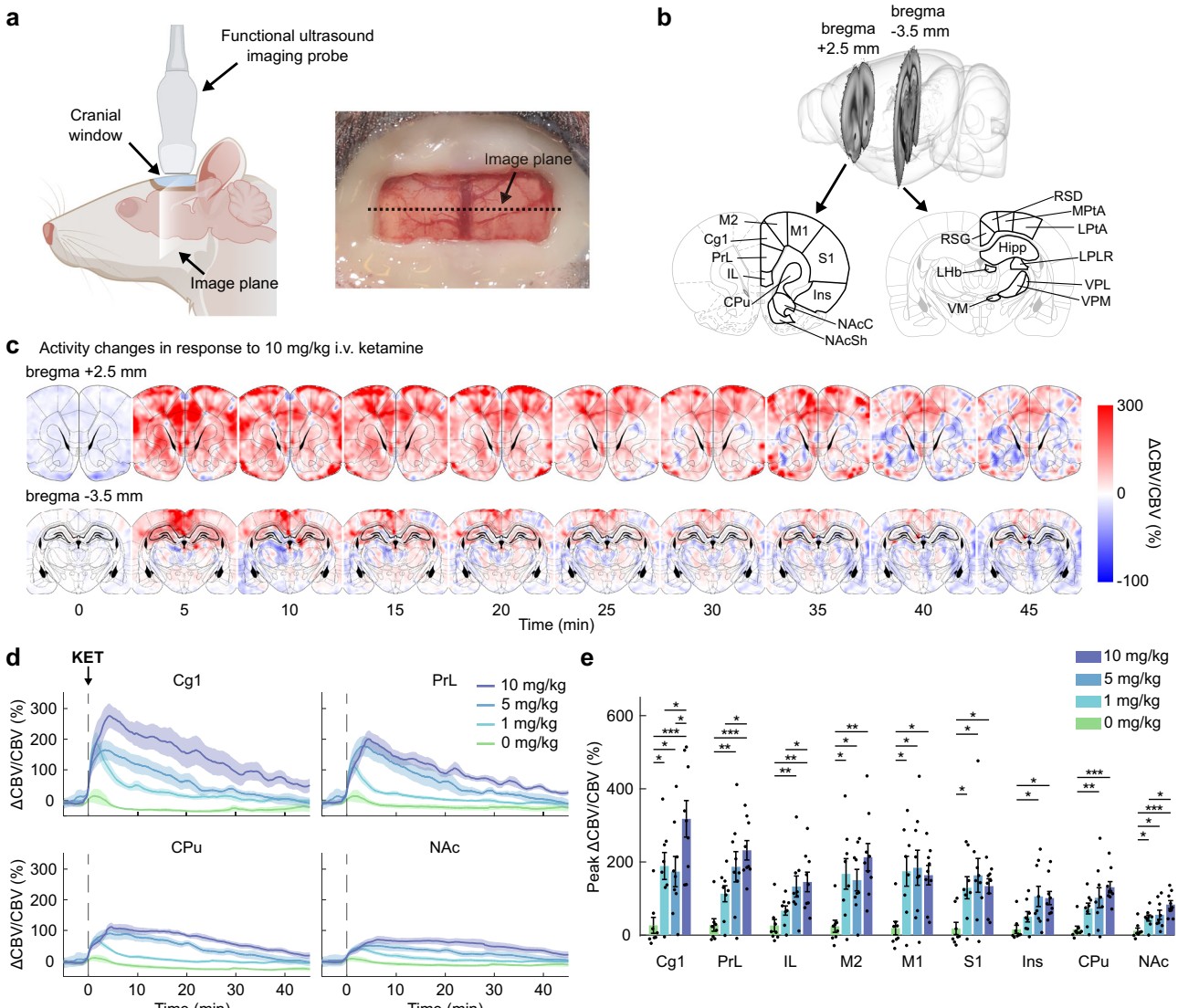

**Fig. 1 | Functional ultrasound imaging of intravenous ketamine administration. a** Schematic representation of the imaging setup. A surgical craniotomy enables ultrasound penetration, and an implanted chronic prosthesis allows imaging over repeated sessions. Drawing created in BioRender. **b** Coronal slices of the rat brain were imaged at bregma +2.5 mm and bregma −3.5 mm. The segmented regions of interest (ROIs) are highlighted on the Paxinos & Watson rat brain atlas[30]. **c** Sequence of cerebral blood volume (CBV) coronal maps at bregma +2.5 mm and bregma −3.5 mm following administration of 10 mg/kg i.v. ketamine. The pixel intensity shows the CBV signals as a normalized difference with a pre-injection baseline (10 min). The time axis was zeroed at the time of ketamine injection. **d** The coronal maps were segmented and the CBV signals were averaged in the relevant ROIs. The plots show CBV time series in response to increasing doses of i.v. ketamine. Solid lines represent the mean values and shaded areas are SEM from $n = 9$ rats/group (10 and 5 mg/kg) or $n = 8$ rats/group (1 and 0 mg/kg). **e** Peak CBV in the segmented ROIs. Two-way mixed-effects ANOVA; within-subjects factor of region,

$F_{3.08,92.55} = 17.82$, $P = 2.33E\text{-}09$; between-subjects factor of dose, $F_{3,30} = 8.26$, $P = 3.74E\text{-}04$; interaction, $F_{9.25,92.55} = 3.38$, $P = 0.001$. Two-tailed unpaired $t$-test, *corrected $P < 0.05$; **$P < 0.01$; ***$P < 0.001$. $n = 9$ rats/group (10 and 5 mg/kg); $n = 8$ rats/group (1 and 0 mg/kg). Data are presented as mean +/− SEM. Source data are provided as a Source Data file. Details on the statistical analyses are provided in Supplementary Table 1. KET: ketamine. Cg1 cingulate area 1, PrL prelimbic cortex, IL infralimbic cortex, M2 secondary motor cortex, M1 primary motor cortex, S1 primary somatosensory cortex, Ins insular cortex, CPu caudate putamen, NAcC nucleus accumbens core, NAcSh nucleus accumbens shell, LPtA lateral parietal association cortex, MPtA medial parietal association cortex, RSG granular retrosplenial cortex, RSD dysgranular retrosplenial cortex, Hipp hippocampus, LHb lateral habenula, LPLR lateral posterior thalamic nucleus, VM ventromedial thalamic nucleus, VPM ventral posteromedial thalamic nucleus, VPL ventral posterolateral thalamic nucleus.

(Fig. 3b, c; corrected $P < 0.05$). Specifically, naltrexone pretreatment decreased ketamine-induced activity in Cg1, primary and secondary motor cortices (M1/2), dorsal striatum (CPu), and nucleus accumbens (NAc), and increased activity in the retrosplenial granular cortex (RSG), lateral habenula (LHb), and lateral posterior thalamic nucleus (LPLR) (Supplementary Movie 2). Interestingly, these effects were only present in male rats, whereas females showed only minor clusters of significant differences between the NTX + KET and VEH + KET treatment conditions (Fig. 3c). These sex-dependent responses were also evident

in the regional CBV time series (Fig. 3d, e). The temporal dynamics of the naltrexone pretreatment effect were transient and region dependent. We observed a biphasic effect where group differences in Cg1, M1/2, CPu, NAcC, and lateral (LPtA) and medial parietal association cortex (MPtA) were mostly limited to the 0−25 min interval, whereas the effect was relatively delayed in RSG, LHb, and LPLR (Supplementary Figs. 4, 5).

We then performed post-hoc statistical analyses on the peak CBV values. Two-way mixed-effects ANOVA stratified by brain region with

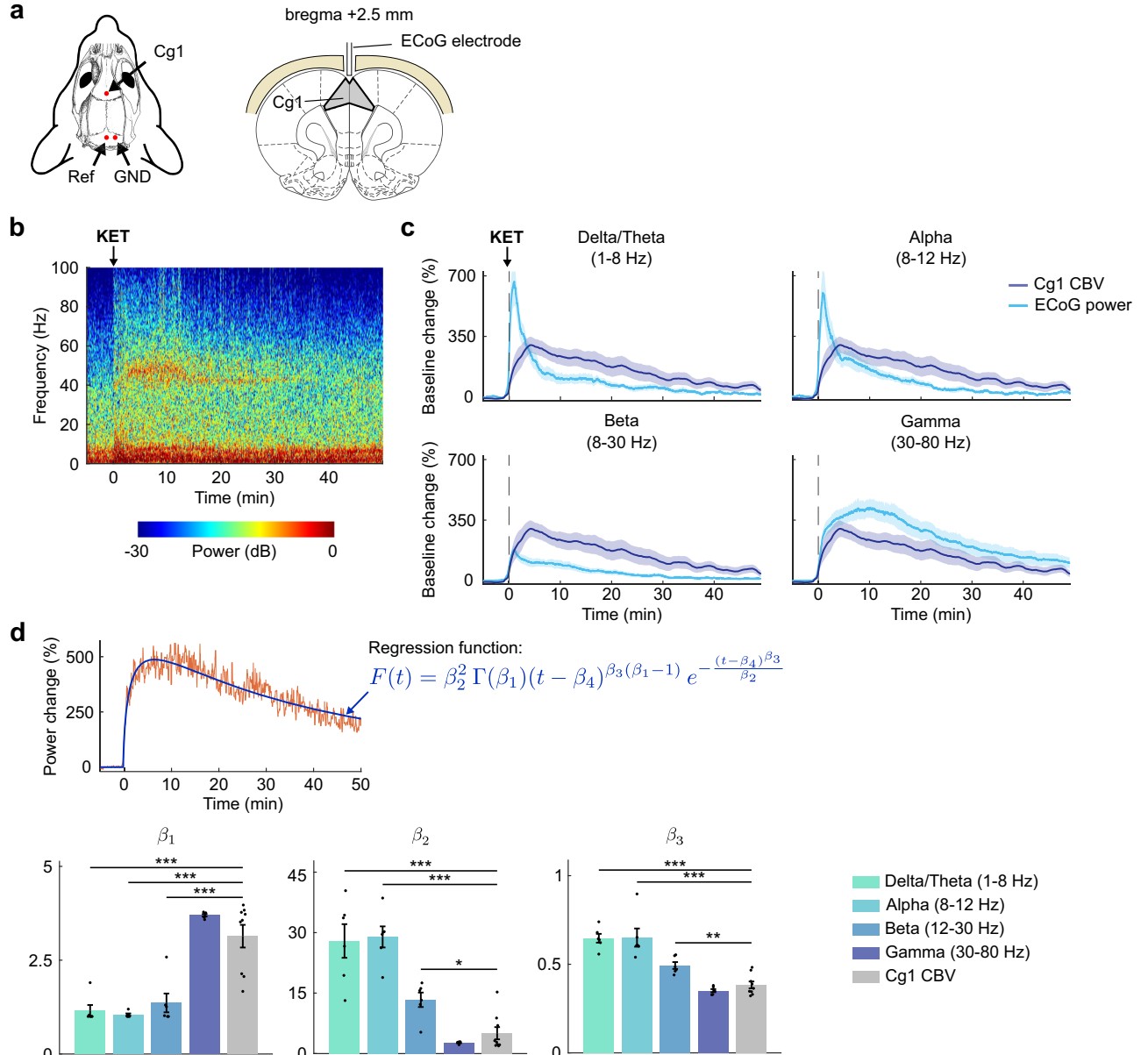

**Fig. 2 | Pharmaco-fUSI closely tracks ketamine-evoked gamma-band power in the prefrontal cortex. a** Schematic representation of the setup for recording intracranial electrocorticography over the Cg1 sub-region of the prefrontal cortex. Coronal slice drawing adapted from the Paxinos & Watson rat brain atlas[30]. **b** Representative spectrogram for i.v. administration of 10 mg/kg ketamine (KET). **c** Time series of normalized electrocorticography (ECoG) power changes in each frequency band and cerebral blood volume (CBV) signal in the Cg1 region. Solid lines represent the mean values and shaded areas are SEM. **d** For each rat, the time series of normalized ECoG power changes and the Cg1 CBV signal were regressed

using a four-parameter gamma-distribution function. The fitted $\beta_1$, $\beta_2$, and $\beta_3$ values were compared between each band and Cg1 CBV time series. A time-delay parameter $\beta_4$ was added to the model to improve the goodness of fit but was not included in the statistical analysis. No significant differences were observed in the least-squares minimization residuals (Supplementary Fig. 3f; $P > 0.11$). Two-tailed unpaired $t$-test comparing each band to CBV, *corrected $P < 0.05$; **$P < 0.01$; ***$P < 0.001$. $n = 6$ rats for ECoG; $n = 9$ rats for CBV. Data are presented as mean +/− SEM. Source data are provided as a Source Data file. Details on the statistical analyses are provided in Supplementary Table 1.

---

between-subjects factor of sex and within-subjects factor of treatment (NTX + KET or VEH + KET) showed a significant effect of sex in LPLR ($F_{1,16} = 6.68$, $P = 0.02$), significant effect of treatment in LHb ($F_{1,16} = 7.42$, $P = 0.015$) and LPLR ($F_{1,16} = 7.79$, $P = 0.013$), and significant sex × treatment interactions in Cg1 ($F_{1,16} = 15.54$, $P = 0.001$), CPu ($F_{1,16} = 7.58$, $P = 0.014$), NAcC ($F_{1,16} = 6.17$, $P = 0.024$), and LPLR ($F_{1,16} = 8.67$, $P = 0.01$) (Supplementary Table 2). In the anterior slice (bregma +2.5 mm), two-way ANOVA stratified by sex with within-subjects factors of brain region and treatment (NTX + KET or VEH + KET) showed a significant effect of region in females ($F_{2.3,18.7} = 31.3$,

$P = 5.20\text{E-}07$) and significant effects of region ($F_{2.1,17} = 16.85$, $P = 7.74\text{E-}05$), treatment ($F_{1,8} = 15.33$, $P = 0.004$), and treatment × region interaction ($F_{2.3,18} = 6.13$, $P = 0.008$) in males. In the posterior slice (bregma −3.5 mm), the same analysis showed a significant effect of region in both female ($F_{1.5,11.9} = 35.73$, $P = 2.18\text{E-}05$) and male rats ($F_{3,24} = 49.27$, $P = 2.08\text{E-}10$) (Supplementary Table 2). Post-hoc pairwise comparisons (two-tailed paired $t$-test) showed significant differences in male rats between the NTX + KET and VEH + KET treatments in Cg1 (corrected $P = 0.024$), M1 ($P = 0.024$), and NAcC ($P = 0.048$) (Fig. 4a, Supplementary Fig. 6a). Group differences in CPu, LHb, and LPLR were only

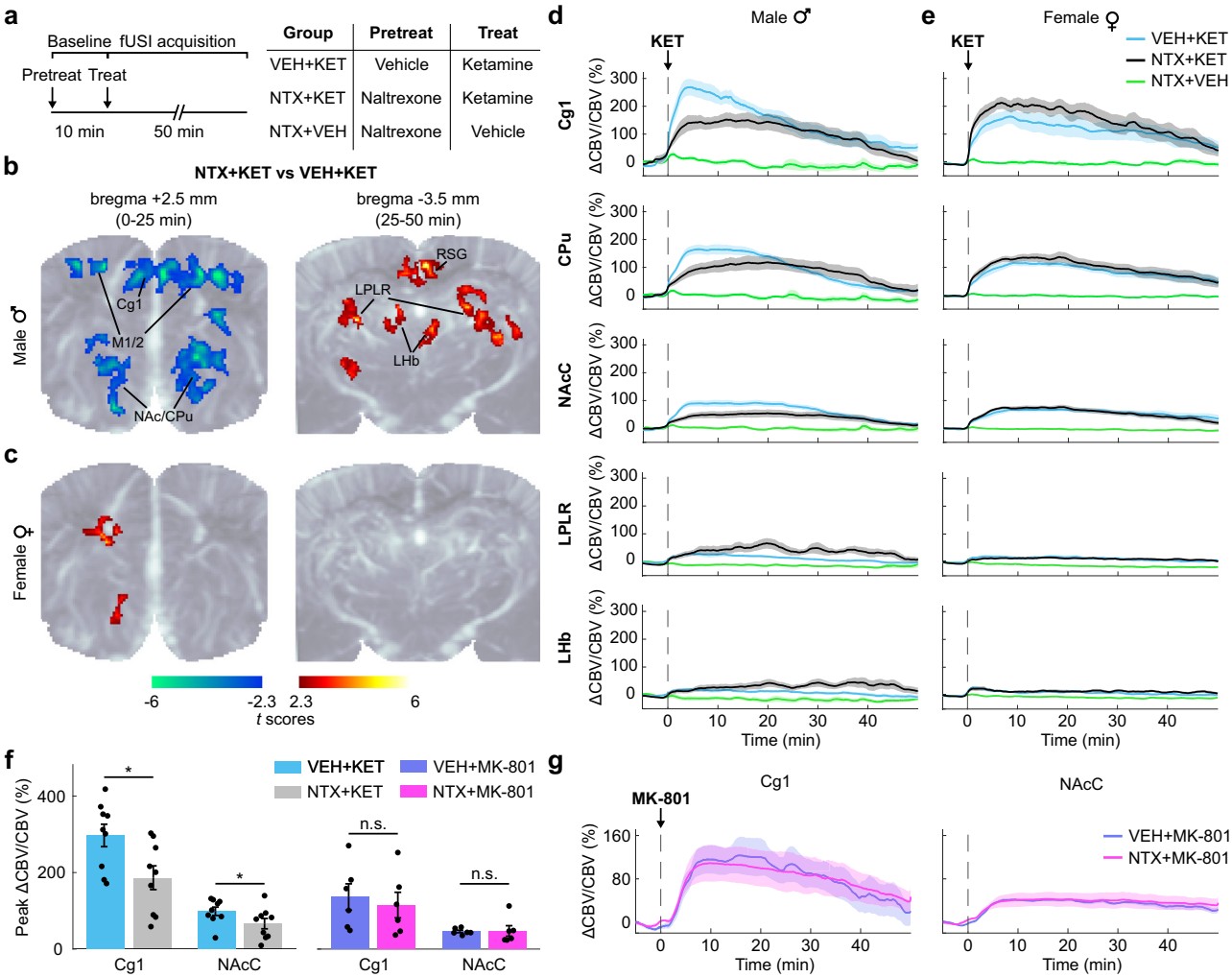

**Fig. 3 | Pharmaco-fUSI reveals a sex-dependence of opioid-mediated effects of ketamine administration. a** Rats received an s.c. injection of either vehicle (VEH; saline) or naltrexone (NTX; 10 mg/kg) followed by ketamine (KET; 10 mg/kg, i.v.) or vehicle after 10 min. Each animal was imaged three times under the treatment conditions of VEH + KET, NTX + KET, and NTX + VEH. **b, c** Functional maps in male (**b**) and female (**c**) rats. The t scores were calculated by contrasting the pixel-wise peak cerebral blood volume (CBV) in the NTX + KET versus VEH + KET groups. Statistically significant clusters are displayed overlaid on a power Doppler template (one cohort of *n* = 9 females and n = 9 males imaged at bregma +2.5 mm; one cohort of *n* = 9 females and *n* = 9 males imaged at bregma −3.5 mm; two-tailed paired *t*-test, corrected *P* < 0.05). In male rats, functional maps show that naltrexone reduced peak activity in M1/2, Cg1, NAc, and CPu, and increased activity in RSG, LHb, and LPRL. There were only minor clusters in female rats. **d**, **e** CBV time series in Cg1,

Cpu, NAcC, RSG, and LHb in male (**d**) and female (**e**) rats. Solid lines represent the mean values and shaded areas are SEM. **f** A different cohort of male rats received an i.v. dose of 0.1 mg/kg MK-801 with naltrexone or vehicle pretreatment. The bar plots display the peak CBV in the Cg1 and NAcC regions. Two-tailed paired *t*-tests (NTX + KET vs VEH + KET and NTX + MK-801 vs VEH + MK-801), corrected P: Cg1 = 0.016 (KET), 0.572 (MK-801); NAcC = 0.024 (KET), 0.922 (MK-801). Hedge's g effect sizes: Cg1 = −1.2 (KET), −0.3 (MK-801); NAcC = −0.97 (KET), 0.04 (MK-801). *n* = 9 male rats (KET groups); *n* = 6 male rats (MK-801 groups). Data are presented as mean +/− SEM. **g** CBV time series in Cg1 and NAcC in male rats receiving MK-801. *n* = 6 male rats. Solid lines represent the mean values and shaded areas are SEM. Source data are provided as a Source Data file. Details on the statistical analyses are provided in Supplementary Table 1.

significant before multiple comparisons correction. No significant region-specific differences were observed between the NTX + KET and VEH + KET treatments in female rats.

To further investigate the sex dependence in the effect of naltrexone pretreatment, we analyzed intra-individual differences in peak CBV between the NTX + KET and VEH + KET treatment conditions. We observed a significant effect of sex (one-way ANOVA, $F_{1,178}$ = 7.52, *P* = 0.007), with specific differences between male and female rats in Cg1 (two-tailed unpaired *t*-test; corrected *P* = 0.012), CPu (*P* = 0.047), and LPLR (*P* = 0.047) (Fig. 4b). Differences in NAcC were significant before multiple comparisons correction. Interestingly, when we compared intra-individual peak CBV differences between the NTX + KET and NTX + VEH groups, we observed a much weaker sex effect (one-way ANOVA, $F_{1,178}$ = 3.22, *P* = 0.075) and no significant regional differences between males and females, suggesting that responses to

ketamine were comparable in males and females when the opioid receptors were blocked (Supplementary Fig. 6b).

In a different cohort of male rats, we performed an orchiectomy (surgical removal of the gonads) to determine whether this sex-dependent effect was driven by endocrine factors rather than developmental sexual dimorphism of the brain[38]. Importantly, the effect of naltrexone pretreatment was completely blocked in orchiectomized males (bregma +2.5 mm slice only; treatment factor: $F_{1,6}$ = 0.31, *P* = 0.6; Fig. 4c, d), suggesting a gating action of testosterone on the opioid-mediated response to ketamine.

To determine if these region and sex-dependent effects were specific to ketamine or caused by variations in the response to naltrexone, we analyzed intra-individual mean CBV differences between the NTX + KET and VEH + KET treatment conditions during the pre-ketamine baseline period (Supplementary Fig. 7). We observed neither

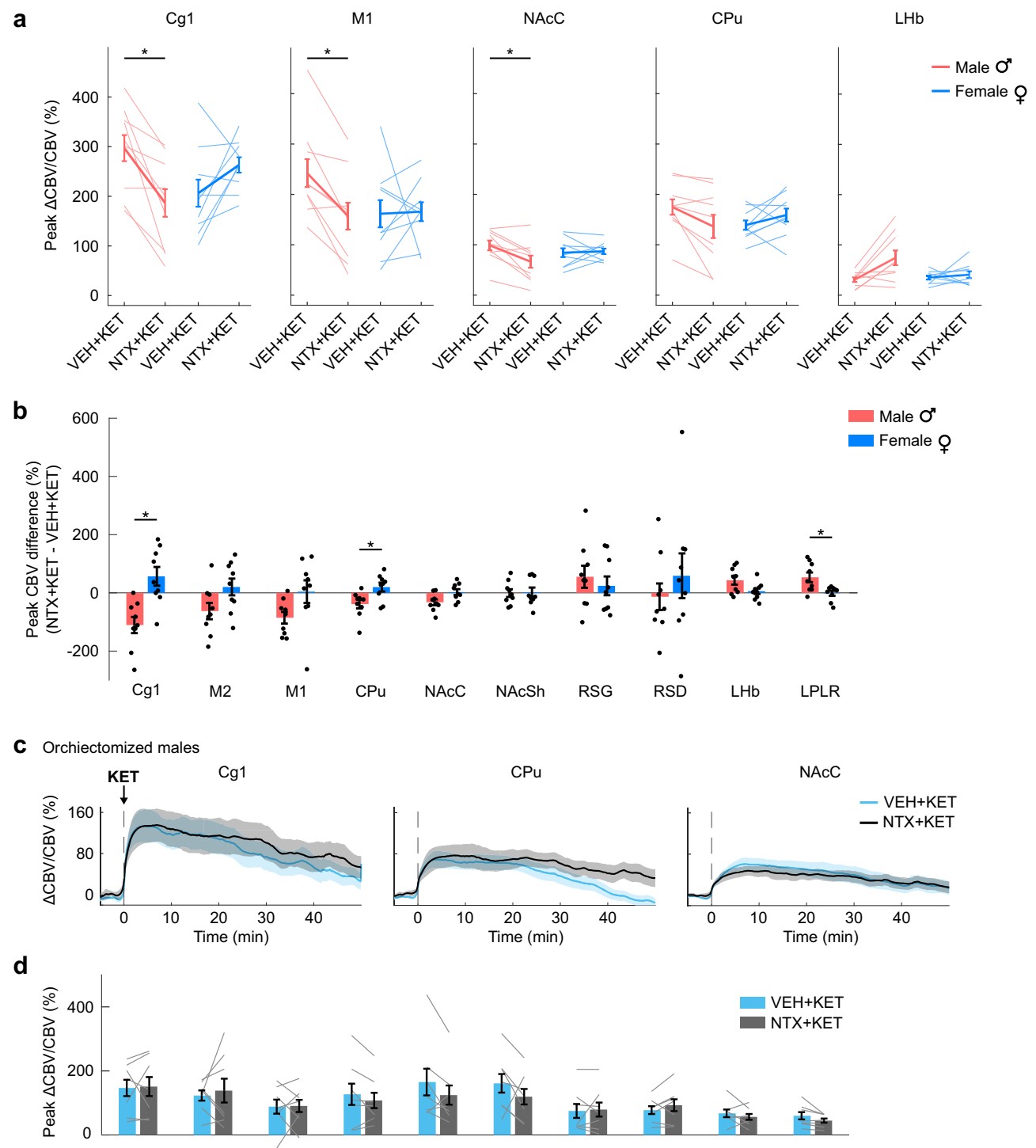

**Fig. 4 | Sex dependence of opioid-mediated effects of subanesthetic ketamine are region-specific. a** Peak cerebral blood volume (CBV) changes in individual male and female rats receiving vehicle or naltrexone pretreatment. Two-tailed paired $t$-test (NTX + KET vs VEH + KET), corrected P: Cg1 = 0.024 (M), 0.23 (F); M1 = 0.024 (M), 0.954 (F); NAcC = 0.048 (M), 0.954 (F); CPu = 0.096 (M), 0.386 (F); LHb = 0.084 (M), 0.77 (F). Group-level differences were significant before multiple comparisons correction in CPu (uncorrected $P$ = 0.032) and LHb ($P$ = 0.021) in male rats. Hedge's g effect sizes (NTX + KET vs VEH + KET): Cg1 = -1.20 (M), 0.53 (F); M1 = -1.28 (M), 0.03 (F); NAcC = −0.97 (M), 0.08 (F); CPu = −0.78 (M), 0.40 (F); LHb = 0.87 (M), 0.17 (F). $n$ = 9 male rats; $n$ = 9 female rats. **b** Peak CBV differences between the NTX + KET and VEH + KET treatments in individual rats were compared between males and females. One-way ANOVA for sex factor, $F_{1,178}$ = 7.52, $P$ = 0.007. Two-tailed unpaired

$t$-test, corrected P: Cg1 = 0.0117; CPu = 0.0473; LPLR = 0.0473. Group-level differences did not pass multiple comparisons correction in NAcC (uncorrected $P$ = 0.0245). Hedge's g effect sizes: Cg1 = 1.77; CPu = 1.24; LPLR = −1.32. **c** A different cohort of male rats received a surgical orchiectomy. CBV time series show that the removal of gonadal hormones completely blocked the opioid-mediated effect of subanesthetic ketamine. $n$ = 7 rats. Solid lines represent the mean values and shaded areas are SEM. **d** Peak CBV in the segmented ROIs. Two-way ANOVA; within-subjects factor of region, $F_{9,54}$ = 3.81, $P$ = 8.9E-04; within-subjects factor of treatment, $F_{1,6}$ = 0.31, $P$ = 0.6; interaction: $F_{9,54}$ = 1.33, $P$ = 0.25. $n$ = 7 rats. Data are presented as mean +/− SEM. Source data are provided as a Source Data file. Details on the statistical analyses are provided in Supplementary Table 1.

a significant effect of sex (one-way ANOVA, $F_{1,178} = 1.83$, $P = 0.178$), nor region-specific differences between sexes (two-tailed unpaired $t$-test, corrected $P > 0.71$), nor sex-specific differences between brain regions (two-tailed paired $t$-test, corrected $P > 0.49$), indicating that the sex dependence in the responses to ketamine was not caused by naltrexone-induced changes in the pre-ketamine baseline but resulted from a more complex pharmacological interaction. Importantly, naltrexone pretreatment produced no significant differences in CBV changes evoked by MK-801, a more selective NMDAR antagonist, at a dose of either 0.1 or 0.25 mg/kg (i.v.) in male rats (bregma +2.5 mm slice only; treatment factor: $F_{1,5} = 0.05$, $P = 0.83$ (0.1 mg/kg) and $F_{1,6} = 0.02$, $P = 0.9$ (0.25 mg/kg); Fig. 3f, g, Supplementary Fig. 8), suggesting that these effects are specific to ketamine. Collectively, our fUSI findings observed during awake acute restraint stress indicate that opioid receptors mediate acute responses specific to subanesthetic ketamine in key brain regions implicated in the pathophysiology of depression and in the processing of reward (e.g., mPFC, NAc, LHb), and that this opioid-dependent effect is critically gated by the presence of male sex hormones.

## Naltrexone suppresses ketamine-induced expression of post-synaptic density protein in male rats

Next, we sought to determine if the opioid-mediated and sex-dependent effects observed in our acute fUSI recordings were reflected in physiological changes at the synaptic level. To this end, we used immunohistochemistry of the postsynaptic density protein PSD-95 in fixed brain slices (Fig. 5a), as an indicator of ketamine-induced cellular structural plasticity. Synapse loss in prefrontal cortical neurons has been identified as a putative neurobiological substrate of depression and other stress-related diseases, and ketamine reverses such synaptic deficits by restoring post-synaptic protein expression and functional spine density[39–41]. Male and female rats received 10 mg/kg ketamine intraperitoneally (i.p.), preceded by naltrexone (10 mg/kg, s.c.) or saline, and were perfused 24 h post infusion. In the quantified PSD-95, a three-way ANOVA with factors of sex, pretreatment (NTX or VEH), and treatment (KET or VEH) revealed significant effects of sex ($F_{1,24} = 33.25$, $P = 6.06\text{E-06}$), pretreatment ($F_{1,24} = 17.21$, $P = 3.62\text{E-04}$), and treatment ($F_{1,24} = 45.49$, $P = 5.63\text{E-07}$), and significant sex × treatment ($F_{1,24} = 17.21$, $P = 3.62\text{E-04}$), sex × pretreatment ($F_{1,24} = 16.38$, $P = 4.68\text{E-04}$), treatment × pretreatment ($F_{1,24} = 42.36$, $P = 9.89\text{E-07}$), and sex × pretreatment × treatment ($F_{1,24} = 15.84$, $P = 5.55\text{E-04}$) interactions. Subanesthetic ketamine increased the expression of PSD-95 in the mPFC of both male (two-tailed unpaired $t$-test; corrected $P = 1.42\text{E-05}$) and female ($P = 0.0071$) rats compared to the saline-injected controls (Fig. 5c, d). In agreement with our imaging findings, naltrexone pretreatment completely blocked ketamine's effect in male rats ($P = 1.86\text{E-}$ 05). Meanwhile, neither males nor females receiving naltrexone alone nor naltrexone and ketamine showed significant differences compared to the saline-injected controls. Importantly, there were no significant differences in the nuclear DAPI staining, considered as a control. Overall, these results indicate that the physiological changes at the synaptic level associated with subanesthetic ketamine administration are indeed significantly blocked with opioid blockade, but only in male rats.

## Ketamine-induced locomotor sensitization is opioid-mediated and sex-dependent

To achieve sustained remission, ketamine therapy typically requires repeated administration over the course of several weeks[42]. Repeated exposure to drugs of abuse in rodent models induces a progressive increase in locomotor behavior (i.e., locomotor sensitization) reflective of neuroadaptations of the mesolimbic dopamine system[43]. Motivated by our observation of altered signaling in mesolimbic structures in our fUSI experiments, we aimed to investigate opioid-mediated sex-dependent effects on behavior in the context of

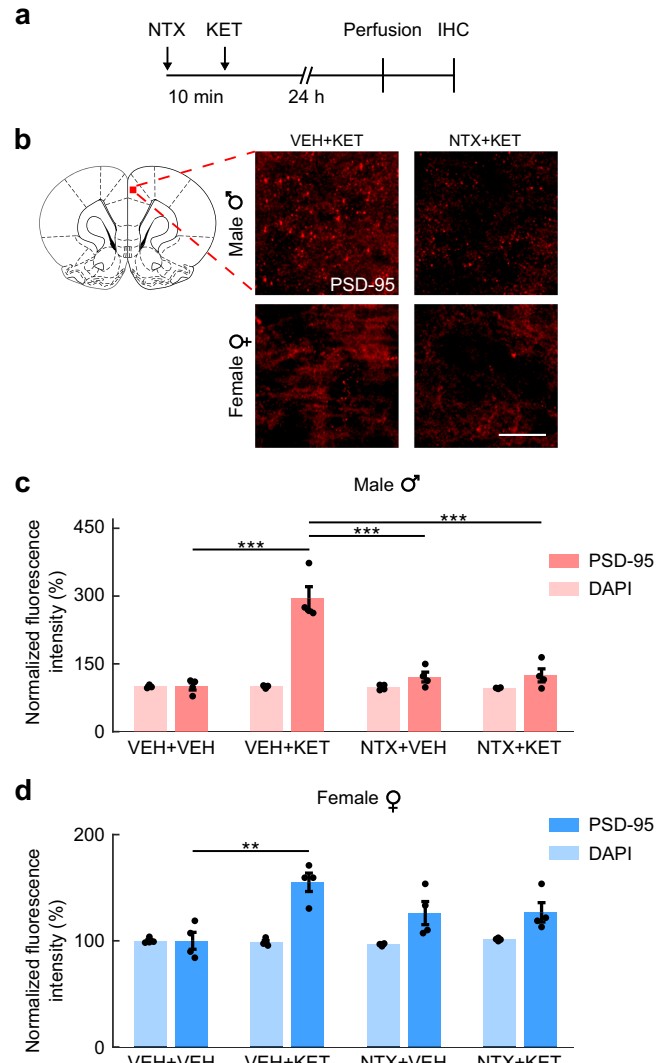

**Fig. 5 | Ketamine infusions increase postsynaptic density in a sex-dependent and opioid-mediated fashion. a** Rats received an s.c. injection of naltrexone (NTX; 10 mg/kg) or vehicle (VEH) followed by ketamine (KET; 10 mg/kg, i.p.) or vehicle after 10 min and were perfused 24 h post ketamine infusion. Brain slices were analyzed by immunohistochemistry (IHC). **b** Representative PSD-95 stains in the rat prefrontal cortex at 20× magnification. Scale bar: 100 µm. Coronal slice drawing adapted from the Paxinos & Watson rat brain atlas[30]. **c** Quantification of prefrontal cortex PSD-95 and DAPI intensity in male rats. Two-tailed unpaired $t$-test (PSD-95), corrected P: VEH + VEH vs VEH + KET = 1.42E-05; VEH + KET vs NTX + VEH = 1.86E-05; VEH + KET vs NTX + KET = 1.86E-05. Hedge's g effect sizes: VEH + VEH vs VEH + KET = 4.36; VEH + KET vs NTX + VEH = −3.75; VEH + KET vs NTX + KET = −3.48. $n = 4$ rats/group. **d** Quantification of prefrontal cortex PSD-95 and DAPI intensity in female rats. Two-tailed unpaired t-test (PSD-95), **corrected P: VEH + VEH vs VEH + KET = 0.00709. Hedge's g effect sizes: VEH + VEH vs VEH + KET = 2.88. $n = 4$ rats/group. Data are presented as mean +/− SEM. Source data are provided as a Source Data file. Details on the statistical analyses are provided in Supplementary Table 1.

repeated ketamine dosing. In a chronic open-field locomotor assay, we pretreated male and female rats with naltrexone (10 mg/kg, s.c.) or saline 10 min before administering subanesthetic ketamine (10 mg/kg, i.p.). Rats were habituated for 2 days with saline only, followed by 4 daily sessions with ketamine with or without naltrexone pretreatment (Fig. 6a). Repeated ketamine administration induced locomotor sensitization in both male and female rats (Fig. 6b–e). Importantly, naltrexone pretreatment completely blocked this effect in males. However, in female rats there was no such sustained blockade of

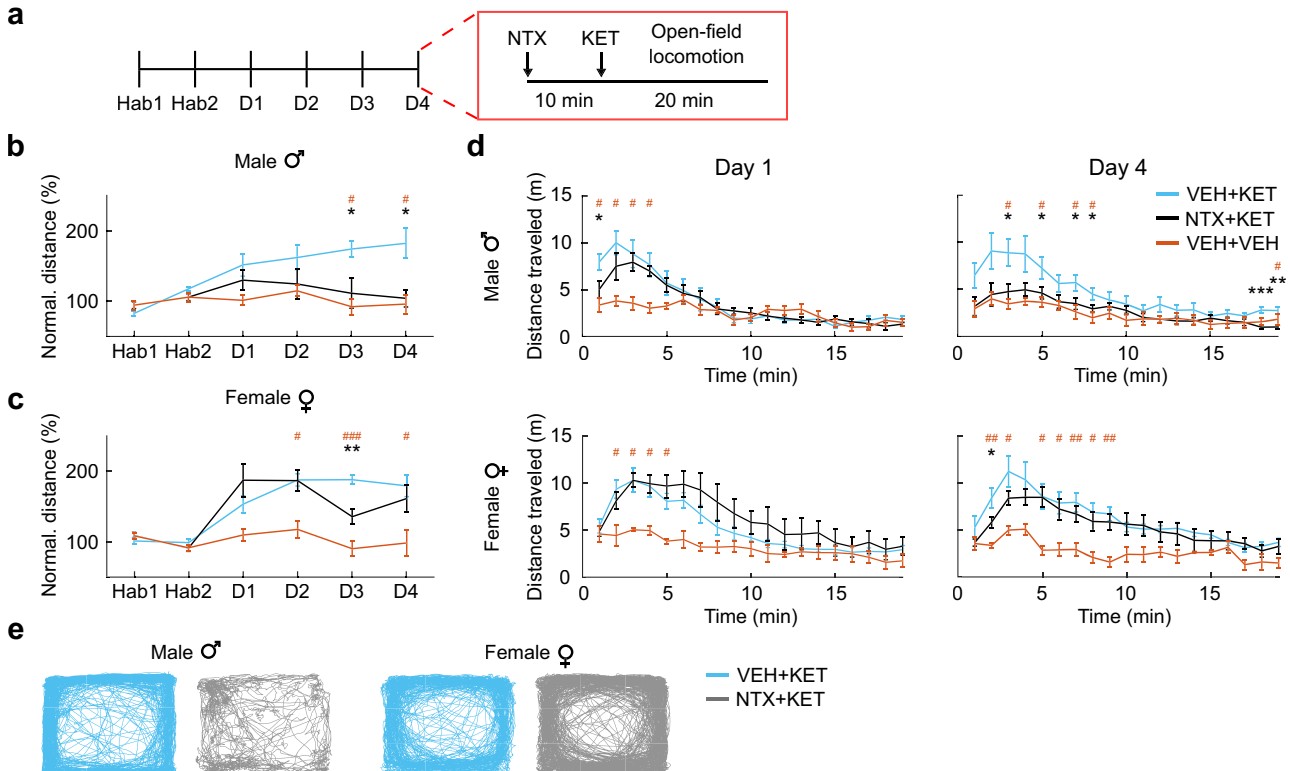

**Fig. 6 | Sex dependence of opioid-mediated behavioral effects of subanesthetic ketamine. a** Locomotor activity was measured for 20 min in an open-field arena. Male and female rats received an s.c. injection of either vehicle (VEH; saline) or naltrexone (NTX; 10 mg/kg) followed by ketamine (KET; 10 mg/kg, i.p.) or vehicle 10 min later on each of four days. All animals were habituated for 2 days (Hab1-2) with vehicle only, followed by 4 daily sessions with ketamine with or without naltrexone pretreatment (D1-4). **b** Total distance traveled by male rats normalized to the mean of Hab1-2. Two-way mixed-effects ANOVA; between-subjects factor of treatment, $F_{2,17} = 5.24$, $P = 0.017$; within-subjects factor of session, $F_{2.74,46.6} = 5.25$, $P = 0.004$; interaction, $F_{5.48,46.6} = 3.40$, $P = 0.009$. VEH + KET vs VEH + VEH: two-tailed unpaired $t$-test, #corrected P < 0.05; Hedge's g effect sizes: D3 = 2.79; D4 = 1.70. NTX + KET vs VEH + KET: two-tailed unpaired $t$-test, *corrected P < 0.05; Hedge's g effect sizes: D3 = −1.25; D4 = −1.49. $n = 8$ male rats/group for VEH + KET and NTX + KET. $n = 4$ male rats for VEH + VEH. **c** Total distance traveled by female rats. Two-way mixed-effects ANOVA; between-subjects factor of treatment,

$F_{2,17} = 9.89$, P = 0.001; within-subjects factor of session, $F_{5,85} = 15.1$, P = 1.35E-10; interaction, $F_{10,85} = 4.16$, P = 0.001. VEH + KET vs VEH + VEH: two-tailed unpaired $t$-test, #corrected P < 0.05; ###P < 0.001; Hedge's g effect sizes: D2 = 2.69; D3 = 4.65; D4 = 1.83. NTX + KET vs VEH + KET: two-tailed unpaired $t$-test, **corrected P < 0.01; Hedge's g effect sizes: D2 = −0.03; D3 = −1.97; D4 = −0.35. $n = 8$ female rats/group for VEH + KET and NTX + KET. $n = 4$ female rats for VEH + VEH. **d** Distance traveled by male and female rats as a function of the time on Day 1 and Day 4. VEH + KET vs VEH + VEH: two-tailed unpaired $t$-test, #corrected P < 0.05; ##P < 0.01. VEH + KET vs NTX + KET: two-tailed unpaired $t$-test, *corrected P < 0.05. $n = 8$ male and female rats/group for VEH + KET and NTX + KET. $n = 4$ male and female rats for VEH + VEH. **e** Representative traces of body position during the open-field session at D4 in male and female rats. Data are presented as mean +/− SEM. Source data are provided as a Source Data file. Details on the statistical analyses are provided in Supplementary Table 1.

ketamine-induced locomotor sensitization; we observed a significant reduction of locomotor activity at Day 3, which was fully reversed at Day 4 (Fig. 6c). Statistical analyses stratified by treatment showed a significant effect of sex (two-way mixed-effects ANOVA; $F_{1,14} = 6.22$, $P = 0.026$), session ($F_{5,70} = 8.82$, $P = 1.59E-06$), and interaction ($F_{5,70} = 3.14$, $P = 0.013$) in rats pretreated with naltrexone, and a significant effect of session ($F_{2.85,39.9} = 29$, $P = 6.66E-10$) in rats pretreated with saline before ketamine administration (Supplementary Table 3). Animals in the control group presented no significant effects. In summary, our results indicate that ketamine produced locomotor sensitization, a marker of mesolimbic dopaminergic adaptation, in both male and female rats, and pretreatment with the opioid receptor antagonist naltrexone completely blocked this effect in male rats only.

### Chronic naltrexone upregulates mu opioid receptors in female rats

We next investigated the molecular mechanisms underlying the opioid-dependent behavioral adaptations induced by chronic dosing of ketamine and naltrexone in male rats. To this end, we performed ex vivo autoradiography with the selective mu opioid receptor (MOR) agonist [3H]DAMGO, following our previous study showing significant

modulation of MOR density in key brain regions with the *(S)*-ketamine isomer[17]. Male and female rats were pretreated with naltrexone (10 mg/kg, s.c.) or vehicle 10 min prior to ketamine (10 mg/kg, i.p.) or vehicle for four consecutive days to mirror our behaviorally relevant dosing protocol (Fig. 6). Twenty-four hours after the last injection the animals were euthanized, and brain slices were incubated with [3H]DAMGO and imaged using a phosphor imager. We have previously shown that repeated *(S)*-ketamine infusions (20 mg/kg/day for 8 days) decreased MOR density in the mPFC, NAc, and thalamus in male and female rats[17]. Curiously, here we did not observe a reduction in MOR density in either males or females following daily injections of 10 mg/kg racemic ketamine (two-tailed unpaired $t$-test, corrected $P > 0.217$) (Fig. 7). However, we found a statistically significant effect of sex in the NAc (two-way ANOVA; $F_{1,40} = 5.12$, $P = 0.029$) (Fig. 7b), and post-hoc analysis in this region revealed statistically significant differences with naltrexone treatment in female rats (both in the NTX + KET vs VEH + KET and NTX + VEH vs VEH + VEH comparisons; two-tailed unpaired $t$-test, corrected $P < 0.036$). In addition, there were statistically significant differences between males and females in the NTX + VEH group in the NAc (two-tailed unpaired $t$-test, corrected $P < 0.039$), and in the NTX + KET group in the CPu (two-tailed unpaired $t$-test, corrected

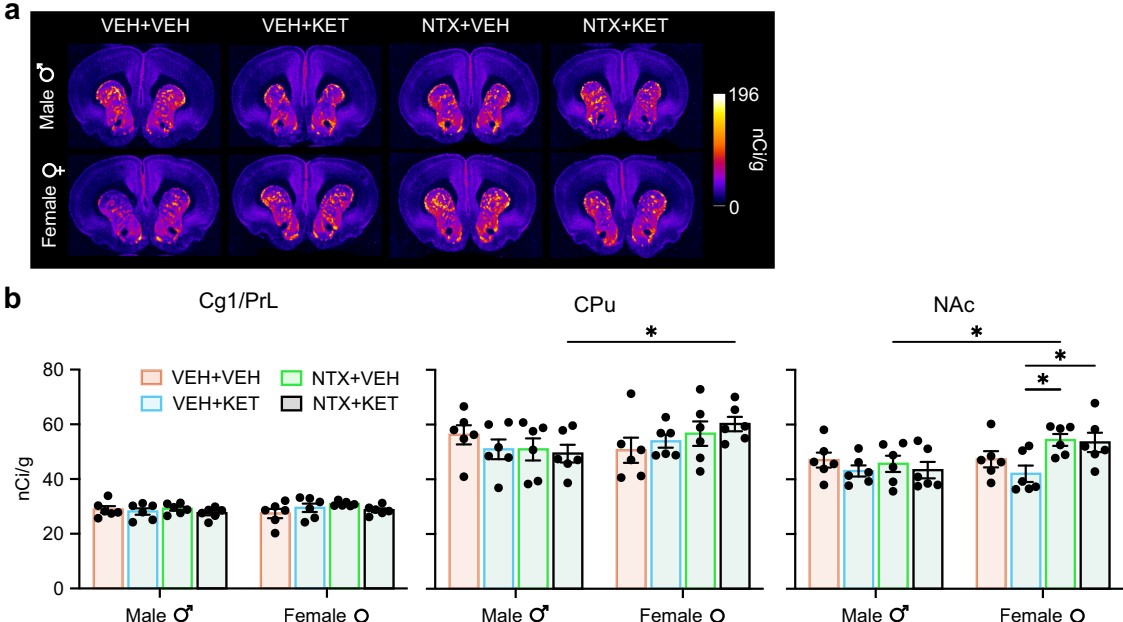

**Fig. 7 | Chronic naltrexone administration increases the density of mu opioid receptors in female rats. a** For four consecutive days, male and female rats were pretreated with naltrexone (10 mg/kg, s.c.) or vehicle 10 min prior to ketamine (10 mg/kg, i.p.) or vehicle and were euthanized 24 h after the last treatment. Autoradiography was performed to quantify [³H]DAMGO binding in the mPFC, CPu, and NAc. The images show representative autoradiography slices. **b** Quantified [³H]DAMGO binding potential in Cg1/PrL, CPu, and NAc. In NAc: two-way ANOVA; sex, $F_{1,40} = 5.12$, $P = 0.029$; treatment, $F_{3,40} = 2.59$, $P = 0.066$; interaction $F_{3,40} = 2.03$, $P = 0.125$. Two-tailed unpaired t-test, *corrected $P < 0.05$. Hedge's g effect sizes: CPu, NTX + KET, male vs female = 1.38; NAc, NTX + VEH, male vs female = 1.26; NAc, female, VEH + KET vs NTX + VEH = 1.79; NAc, female, VEH + KET vs NTX + KET = 1.33. $n = 6$ male *and* female rats/group. Data are presented as mean +/− SEM. Source data are provided as a Source Data file. Details on the statistical analyses are provided in Supplementary Table 1.

$P < 0.027$). Our results suggest that repeated naltrexone administration increased MOR density in the NAc selectively in female rats.

This effect could potentially explain why ketamine-induced locomotor sensitization was not blocked by naltrexone pretreatment in females: while naltrexone fully blocked MORs in male rats preventing behavioral sensitization, females had increased opioid system capacitance to increase MOR density in response to naltrexone exposure, and as a result, racemic ketamine could still actuate a sufficient number of receptors in female rats to elicit a sensitized locomotor effect. Importantly, although we did not find significantly decreased MOR density from 10 mg/kg racemic ketamine (equivalent to 5 mg/kg *(S)*-ketamine) administration compared to vehicle, we observed a mean reduction of 9.8% in [³H]DAMGO binding in the NAc when we aggregated male and female rats. For comparison, in our previous study the binding potential in the NAc was reduced by 29.5% in male and female rats after 8 daily i.v. infusions of 20 mg/kg *(S)*-ketamine[17]. While it is difficult to directly compare these results due to the different dosing regimens, it is worth noting that in our prior study the cumulative dose of *(S)*-ketamine was 8x higher (160 mg/kg vs 20 mg/kg assuming 50% *(S)*-ketamine isomer in the racemic formulation). Therefore, considering that *(S)*- but not *(R)*-ketamine binds MORs[44], it was expected that we would observe a lesser reduction in MOR density in this current experiment.

## Discussion

Our results indicate that opioid receptors mediate, at least partly, the neural activity changes elicited by a subanesthetic dose of racemic ketamine as measured by pharmaco-fUSI in the acute restraint rat model of stress. Although ketamine's mechanism of action in this context is not fully clear, one prevailing hypothesis is that subanesthetic ketamine causes NMDAR-mediated inhibition of fast-spiking gamma aminobutyric acid (GABA)-ergic interneurons in the mPFC, with a consequent glutamate surge and disinhibition of pyramidal cells, resulting in a net cortical excitatory effect[45,46]. NMDAR-independent pathways have also been reported[9], along with inhibition of different types of cortical interneurons[47]. We demonstrate here that opioid receptors also play a critical role in subanesthetic ketamine's action in the mPFC and other related cortical and subcortical regions, and importantly, that these opioid-mediated effects are sex-dependent.

In the Cg1 sub-region of the mPFC, we show that ketamine-evoked fUSI signals closely track ECoG power changes in the gamma band, confirming the neural basis of these fUSI signals. Gamma-band oscillations are regulated by NMDAR signaling in fast-spiking interneurons[48]. These neurons send wide-spread inhibitory projections to excitatory pyramidal cells and are important for cognitive functions such as attention, learning, and memory[49]. Acute ketamine administration leads to robust, dose-dependent gamma-band power enhancement throughout the neocortex and subcortical regions[32,50], which coheres with our fUSI observations. Notably, a high correlation between fUSI signals and gamma (30–90 Hz) and high gamma-band (110–170 Hz) local field potentials has been previously reported in the visual cortex and hippocampus of mice[33].

Ketamine's opioid-mediated responses were observed in the mPFC, NAc, and LHb, neural structures implicated in processing of reward and cognitive functions relevant for the pathophysiology of depression and other psychiatric diseases[51–54]. Notably, many of these regions have been previously reported to show increased early gene (cFos) expression upon ketamine and psylocibin administration[55]. Our results are also in agreement with prior reports of ketamine directly recruiting opioid receptors in the mPFC[16] and LHb[14]. Opioid signaling in the mPFC is consistent with ketamine's antinociceptive and analgesic action[56], as this region executes descending pain control via functional connections with the periaqueductal gray[57], and activation of opioid receptors decreases LHb activity[58]. The observed opioid-mediated response to ketamine in the NAc, a critical site for reward

processing[59,60] and closely linked to the anhedonic symptoms of depression[52], may originate either from direct opioidergic action at this site[61] or via downstream projections from the mPFC[62]. Our recent study showed that (S)-ketamine occupies mu opioid receptors in the NAc and elicits opioid-mediated activation of this region[17]. Racemic ketamine was also found to induce dopamine transients in the NAc similar to those evoked by cocaine[63], although it was concluded to have limited addiction potential.

Interestingly, the opioid-mediated effects that we observed were critically dependent on the presence of male sex hormones, as modulation of ketamine-evoked responses by opioid blockade was not evident in female subjects and was fully reversed by surgical removal of the male gonads. These acute observations were reflected in downstream changes in postsynaptic density in the mPFC, a putative biomarker of the antidepressant action of ketamine[5]. Importantly, repeated dosing of subanesthetic ketamine induced behavioral sensitization, a robust measure of the neurobehavioral adaptations caused by repeated exposure to drugs with abuse liability[64], in both male and female rats. Blocking the opioid receptors also suppressed this behavioral effect, but only in male subjects. Finally, we showed that the lack of ketamine-evoked behavioral adaptations in females may be explained by a compensatory effect of repeated naltrexone infusions, which increased MOR density in the NAc in female rats only. Previous studies reported upregulation of MORs following chronic administration of naltrexone or naloxone[65,66]. Here we add to this knowledge by highlighting a clear sex dependence of these effects. Synthesizing our current results with those in the literature, including the differential pharmacokinetics of ketamine in male and female humans and rodents[67], our findings suggest that at doses that we observed to be behaviorally relevant, ketamine does indeed act partly through opioid pathways to induce varied physiologic, cellular, and behavioral-level effects, but that female rats have a greater capacity to upregulate opioid pathways and activate a compensatory mechanism in the setting of opioid blockade. We posit that this differential capacitance for modulating opioid signaling between varied subject populations may account for the heterogeneity of results across clinical trials that have attempted to investigate the role of the opioid pathway in the affective effects of subanesthetic ketamine[10-13].

It is important to note that these results from rats would need to be explicitly verified in clinical trials before drawing meaningful conclusions with respect to clinical treatments using subanesthetic ketamine. With regards to a sex-dependence of the effects of ketamine, proper comparisons of ketamine's therapeutic and adverse effects in male and female patients are lacking, possibly due to the limited statistical power for such subgroup analysis in current clinical trials, although sex differences have previously been reported at an anecdotal level[68]. Sex-dependent ketamine and norketamine pharmacokinetics have been observed[67], and diverging correlations between treatment outcomes and depression-related inflammatory cytokines have been reported in male and female subjects[69]. Moreover, preclinical investigations showed sex differences in the pharmacokinetics of ketamine and its metabolites, as well as in behavioral and physiological readouts[9,67,70]. Our results elucidate and emphasize both the sex-dependent and opioid-based mechanisms underlying the actions of subanesthetic ketamine. The timeliness for such investigation is made more urgent by the current widespread administration of ketamine, in both its racemic and isomer-specific forms, in patients with treatment-resistant depression and other affective disorders. Indeed, it is possible that the current heterogeneity in the clinical and preclinical findings and the associated controversy surrounding the potential opioid dependence of ketamine's clinical effects could reflect the existence of explanatory demographic-based biological variables, such as sex. As these current results regarding a sex dependence in rats were revealed only with opioid blockade, they underscore the need for future clinical trials of the interaction of ketamine and opioids to be sufficiently powered to detect sex-based differences. To this end, we point out that in one of the recent studies reporting no effects of opioid receptor blockade on ketamine's antidepressant action[13], the single subject receiving naltrexone concurrently with ketamine infusions was a female participant. While several factors of that study prevent definitive deductions, including its limited sample size and the investigation in a population of substance abuse and pain patients with likely differential status from the general population in terms of opioid receptor density, our results appear to be in agreement with these findings and may help explain the conflicting observations in the clinical literature.

There are several limitations to our study. First, as we have noted, these results from rodents would need explicit verification in controlled clinical trials to draw meaningful conclusions for clinical treatments using ketamine. Second, subanesthetic ketamine causes transient cardiovascular effects in both humans and rats[3,71], which may confound the interpretation of our fUSI results. However, the dynamics of ketamine's activity and the effect of naltrexone pretreatment varied substantially between brain regions and throughout the scanning time, with opposing effects in different brain regions, suggesting that a central cardiovascular modulatory effect of ketamine is unlikely to account for our brain region-specific results. In addition, we show that ketamine-evoked CBV signals in the Cg1 sub-region of the mPFC closely tracked electrophysiological changes in the gamma-band measured over this region, supporting that our fUSI results correlate more to changes in neural activity than a central cardiovascular modulation. Also, we did not see a correlation of the imaging results with the baseline vascular density of each region. Moreover, previous studies also showed similar patterns of neural activity with complementary modalities including pharmacological magnetic resonance imaging[72] and [18F]-fluorodeoxyglucose positron emission tomography[16]. Therefore, we consider that our observed fUSI findings were unlikely to reflect a ketamine-induced global cardiovascular modulation. Third, in female rats we did not control for the estrous cycle at the time of injection[73]. However, the physiological effects of ketamine have not been shown to be dependent on the estrous phase (diestrus or proestrus)[40], and surgical ovariectomy did not alter the plasma levels of ketamine and its metabolites in mice[67]. Moreover, our randomized study design should mitigate any confounds related to the estrous phase, as female rats in each treatment group were likely imaged during different phases of the estrous cycle.

In summary, our results establish that opioid blockade can modulate neural activity, cellular physiologic, and behavioral changes induced by subanesthetic ketamine, but only in male rats. Therefore, it is imperative that future clinical trials focus on sex as a biological variable in assessing the affective responses to subanesthetic ketamine, including its antidepressant efficacy, especially with respect to potential abuse liability or withdrawal type responses upon discontinuation[74,75]. Excitingly, our regional mapping may inform and guide future studies with ultrasound-mediated interventions for focal delivery of ketamine[76,77].

## Methods
### Animals
All animal procedures were approved by the Institutional Animal Care and Use Committee at Stanford University and at the National Institute on Drug Abuse. Male and female Long Evans rats (Charles River Laboratories) were used in the experiments. All animals were 9-10 weeks old and weighed $278 \pm 40$ g (mean ± s.d.) when they entered the study. Animals had *ad libitum* access to water and food for the entire duration of the experimental protocols. Rats were housed in a temperature-controlled vivarium on a 12-h light-dark cycle (lights on at 7 AM; lights off at 7 PM) and were acclimated to their home cage for one week before experimentation. In case of surgical procedures, the animals were singly housed following the surgery.

## Drugs

Naltrexone hydrochloride (Sigma for the autoradiography experiment; Tocris Bioscience for all the other experiments) was suspended in 0.9% sterile saline to obtain a 10 mg/mL solution. MK-801 (Tocris Bioscience) was suspended in 0.9% sterile saline to obtain 0.1 and 0.25 mg/mL solutions. Ketamine hydrochloride (Covertus for the autoradiography experiment; Dechra Veterinary Products for all the other experiments) was diluted in 0.9% sterile saline to obtain 1, 5, and 10 mg/mL solutions. All drugs were administered in a bolus injected volume of 1 mL/kg.

## Surgical procedures

**Craniotomy.** Rats received a bilateral surgical craniotomy and chronic prosthesis implantation as previously reported[26]. Briefly, animals were anesthetized with 3.5% isoflurane in 100% oxygen, and anesthesia was maintained with 1.5% isoflurane. The incision region was prepared by shaving the skin using a depilatory cream. Rats were then placed in a stereotaxic frame for head fixation and orientation. Body temperature was maintained at 37 °C by a warming pad with rectal probe monitoring (RightTemp Jr.; Kent Scientific). Heart rate and arterial oxygen saturation were monitored by a pulse oximeter (MouseStat Jr.; Kent Scientific). Anti-inflammatory (dexamethasone, 1 mg/kg; i.p.) was administered to prevent brain swelling and inflammation. The incision site was disinfected by applying alternating povidone-iodine and 75% EtOH, and a skin incision was performed. The bone was cleaned with 75% EtOH and a window (5 mm AP × 10 mm ML centered at bregma +2.5 mm) was marked on the skull with a surgical pen. The bone around the window was pretreated using a bonding agent (iBOND Total Etch; Kulzer). We then cut parietal and frontal bone fragments using a handheld high-speed drill with a 0.7 mm drill bit (Fine Science Tools). We gently removed the bone flaps paying attention to avoid damaging the dura mater, and sealed with dental cement (Tetric EvoFlow; Ivoclar Vivadent) a 125-μm polymethylpentene film covering the cranial window. The space between the dura and the prosthesis was filled with 0.9% sterile saline. A dose of 0.5 mg/kg Buprenex SR was administered subcutaneously for analgesia. The animals were allowed to recover for 1 week before the first imaging session.

**Electrode implantation.** Male rats were prepared for surgery as described above. After the skull was exposed, cleaned, and pretreated with bonding agent, we drilled burr holes (0.7-mm drill bit; Fine Science Tools) using a handheld high-speed drill for electrode implantation. A PFA-coated 100-μm stainless steel wire (A-M Systems) was used to create electrical contacts with the cerebral cortex at AP 2.5 mm - ML 0 mm (Cg1), AP -10 mm - ML 0 mm (reference), and AP -10 mm - ML −2.5 mm (ground). Dental cement (Tetric EvoFlow; Ivoclar Vivadent) was used to secure the electrodes. A dose of 0.5 mg/kg Buprenex SR was administered subcutaneously for analgesia, and the animals were allowed to recover for 1 week before the recording session.

**Orchiectomy.** To assess the effect of sex hormones, adult male rats were orchiectomized following previously published protocols[78]. The animals were anesthetized with 3.5% isoflurane in 100% oxygen, and anesthesia was maintained with 1.5% isoflurane. The incision site was prepared by shaving and disinfecting the skin as described above. An incision of about 10 mm was made on the ventral side of scrotum along the midline. The testicular content was exposed, the vas deferens and blood vessels were clamped to prevent bleeding, and the testicles were removed. The incision was then closed with monofilament sutures. We waited for 10 days before experimentation to allow for recovery and testosterone washout.

## Pharmaco-functional ultrasound imaging

**Ultrasound system and power Doppler processing.** A Vantage 256 research scanner (Verasonics Inc.) was connected to a linear array transducer (Vermon; 128 elements, lateral pitch of 100 μm) operating at a 15-MHz center frequency. The imaging probe was housed in a custom 3-D printed holder mounted on a motorized positioning system. For acoustic coupling, we used ultrasound gel that was centrifuged to remove air bubbles. The imaging sequence consisted of five tilted plane waves (-6°, -3°, 0°, 3°, 6°) emitted with a pulse repetition frequency of 19 kHz. Two plane waves were averaged for each angle to increase the signal-to-noise ratio. We acquired data for 200 compound frames at a rate of 1 kHz, and the frames were beamformed in a regular grid of pixels with in-plane resolution of 100 μm × 100 μm. Beamforming was performed in real-time in an NVIDIA Titan RTX using a GPU beamformer[79].

Sequences of 200 compound ultrasound frames were processed offline in MATLAB (MathWorks, Inc.) for clutter filtration and power Doppler computation. To eliminate the Doppler signal component originating from the stationary tissue, we used a 5th-order temporal high-pass Butterworth filter with a cutoff frequency of 40 Hz and a singular value decomposition filter that eliminates the first singular value[80]. The power Doppler intensity at each pixel was calculated by squaring and averaging the filtered Doppler signals. The final power Doppler frame rate was 1 frame/s.

**Imaging session.** At the beginning of each imaging session, rats were briefly anesthetized with isoflurane and a catheter was placed in the tail vein for vascular access. While under anesthesia, animals were placed in a plastic restraint cone (Stoelting Co.) and positioned in a custom head-restraining apparatus[81]. Oxygen was flowed through the nose cone to prevent hypoxia. The ultrasound probe was positioned over the slice of interest. The relevant brain atlas slice was plotted overlaid on the real-time power Doppler images to facilitate accurate probe positioning based on vascular landmarks (Supplementary Fig. 1). With the animal in the imaging apparatus, we waited for 30–45 min before data acquisition to allow for complete isoflurane clearance. An s.c. injection of naltrexone or vehicle was performed, followed by an i.v. injection of drug (ketamine or MK-801) or vehicle after 10 min. After the i.v. injection, the catheter was flushed with 200 μL of sterile saline. We acquired data continuously for 50 min following drug administration.

**Functional ultrasound data pre-processing.** To prevent motion artifacts in the processed CBV signals, translational and rotational movements were corrected by applying a motion correction algorithm to the image time series (Supplementary Fig. 1). For each acquisition, a power Doppler template was calculated via median filtering of the first 500 images. Then, all power Doppler frames from the same acquisition were registered to the template using a rigid transformation that included rotations, translations, and cubic interpolations. A filter was used to remove registered data frames affected by excessive motion or other artifacts. This filter was adapted from previously published code[82]. Each power Doppler dataset was then manually registered to the relevant slice of the Paxinos & Watson rat brain atlas[30] (at bregma +2.5 mm or bregma −3.5 mm).

**Cerebrovascular time series.** The pixel-wise relative CBV signal was calculated as the normalized difference with a baseline (i.e., $\Delta CBV/CBV = (CBV_t - CBV_0) / CBV_0$). For each acquisition, the baseline was calculated by averaging 10 min of power Doppler data immediately before drug administration. The regional time series were computed by spatially averaging the pixel $\Delta CBV/CBV$ signals in the relevant segmented ROIs in each brain slice (Fig. 1a). Time series were time-locked to the time of ketamine administration.

**Functional maps.** To assess the effect of ketamine administration and naltrexone pretreatment, we used an approach similar to direct pharmaco-fMRI[72], where we used pixel-wise statistical inference to

analyze group-level differences in peak CBV signal. For each pre-processed power Doppler acquisition, an image was created by calculating the temporal CBV peak at each spatial location. Peak CBV images were registered to the template atlas space by performing a rigid transformation, and $t$ scores were calculated for the contrasted groups (NTX + KET vs VEH + KET; two-tailed paired $t$ test). Thresholded $t$ scores were corrected for multiple comparisons across each slice using a cluster-size threshold of 34 contiguous pixels. The threshold was determined via Monte Carlo simulations using the 3dClustSim program of the AFNI library[83] to obtain an overall cluster $P < 0.05$, family-wise error rate corrected. Color-coded functional maps were displayed overlaid on a power Doppler template to enable a visual comparison of the analyzed groups.

## Electrocorticography

**Recording.** At the beginning of the recording session, rats were briefly anesthetized with isoflurane and a catheter was placed in the tail vein for vascular access. While under anesthesia, animals were placed in a plastic restraint cone (Stoelting Co.) and positioned in a custom head-restraining apparatus[81]. Oxygen was flowed through the nose cone to prevent hypoxia. Electrocorticography recording was performed with an 8 Channel Cyton Biosensing Board (OpenBCI) using the OpenBCI GUI at a sampling frequency of 500 Hz. With the animal in the restraint, we waited for 30–45 min before data acquisition to allow for complete isoflurane clearance. After a baseline acquisition (10 min), an i.v. injection of drug ketamine (10 mg/kg or 1 mg/kg) was performed, and the catheter was flushed with 200 μL of sterile saline. We acquired data continuously for 50 min following ketamine administration.

**Processing.** Raw ECoG traces were processed using a 5th-order Butterworth filter with cutoff frequencies 1-100 Hz. A short-time Fourier transform was computed in 1-s nonoverlapping temporal segments, and the power of the resulting spectrogram was calculated. The spectral power was then averaged in each band (delta/theta: 1-8 HZ; alpha: 8-12 Hz; beta: 12-30 Hz; gamma: 30–80 Hz) and normalized to the mean power of the baseline period (10-min pre-ketamine) to compute the time series. The time series were time-locked to the time of ketamine administration, and a median temporal filter was applied with a kernel of 15 s for smoothing.

We regressed the ECoG time series in each spectral band and the Cg1 CBV signal using a single-gamma distribution function (Fig. 2c) with four β parameters. The fitting was performed using the 'lsqcurvefit' function in Matlab and iteratively minimized the sum of the squared residuals between the target signal and the fitted curve. The initial β values were (2, 10, 0.5, 0). Stopping criteria were gradient step tolerance and function tolerance of 1E-15 or max 1E4 function evaluation.

The regressed β values were compared between the ECoG bands and the Cg1 CBV time series. A time-delay parameter ($\beta_4$) was included to account for potential uncertainties in the ketamine injection time and to improve the goodness of fit. This parameter was not included in the statistical analysis. All the filtering and regression was performed in Matlab using custom-built scripts.

## Postsynaptic density protein PSD-95

**Drug administration and immunohistochemistry.** Rats were administered an s.c. injection of 10 mg/kg naltrexone or vehicle. After 10 min, an i.p. injection of 10 mg/mg ketamine or vehicle was performed, and the animals were returned to their home cage. After 24 h post-ketamine, the animals were anesthetized with isoflurane (5%) and transcardially perfused with 1x phosphate-buffered saline (PBS) followed by 4% paraformaldehyde (PFA) diluted in PBS. Brains were extracted and fixed overnight in 4% PFA, then subsequently washed in PBS and frozen in embedding medium. Coronal sections 40 μm thickness were cut on a CM1800 Cryostat (Leica Microsystems),

transferred to tissue storage solution (30% sucrose and 30% ethylene glycol in 0.1 M PB), and stored at −20 °C until immunohistochemical processing. Four tissue sections per rat (from Bregma +2.7 to +1.2; one 40 μm section every 400 μm) were selected for PSD-95 and DAPI labeling. Floating sections were rinsed with PBS then blocked with 4% normal goat serum and 0.3% Triton-X 100 diluted in PBS. Sections were then incubated overnight at 4 °C in primary antibody, rabbit monoclonal to PSD95 (ab238135; Abcam). Following incubation, sections were rinsed in PBS and incubated in secondary antibody, goat-anti-rabbit Alexa Fluor 555 (Invitrogen) at 1:500 for 2 h. Sections were then mounted on super-plus glass slides (VWR), airdried in the dark, and cover-slipped with hard-set mounting medium containing DAPI (Vector Labs).

**Microscopy and image analysis.** Images for PSD95-stained sections were acquired on a Keyence BZ-X800 fluorescence microscope (Keyence Corp.). Acquisition settings remained strictly constant between all images acquired at the same magnification. Specific ROIs were chosen to sample the mPFC at approximately the infralimbic, pre-limbic, and cingulate area 1. High-resolution $z$-stacks of each ROI were acquired using a 40× magnification with a step size of 0.4 μm and total depth of 6 μm. Each $Z$-stack image set was merged and analyzed with BZ-X Advanced Analysis Software (Keyence Corp.). Signals above thresholded background were used for manual ROI segmentation to calculate the area of mean fluorescent signal intensity of each ROI, averaged across the four sections collected per animal. Mean fluorescent intensity is reported in arbitrary units.

## Locomotor sensitization

All behavioral tests were performed in an environmentally controlled room. Open-field locomotor activity was recorded in a custom-built white Plexiglas apparatus (90 cm × 90 cm × 40 cm) divided in four equal compartments. Videos were collected for batches of 4 animals using an overheard camera placed at the center of the field. Animals in each batch were randomized for sex and treatment group. Prior to the behavioral tests, rats were handled for 3 days to acclimate to the experimenter and reduce stress. Then, locomotor activity was recorded for a total of 6 days. In the first two habituation days (HAB1/2), rats received an s.c. injection of vehicle and were then returned to their home cage. After 10 min, rats received an i.p. injection of vehicle and were immediately placed at the center of the arena, where they were allowed to freely explore for 20 min while locomotor activity was recorded. In the following 4 days (D1/4), rats were administered an s.c. injection of vehicle or naltrexone (10 mg/kg) and returned to their home cage. After 10 min, rats received an i.p. injection of ketamine (10 mg/kg) and were placed at the center of the arena while locomotion was recorded. Animals in the control group (VEH + VEH) continued to receive vehicle injections for the entire duration of the experiment. The compartments were thoroughly cleaned with Virkon between each recording session to control for scent-related confounds. White noise (65 dB) was played during the sessions to attenuate any external noise. Both a male (TDI) and a female (SNE) experimenter conducted the tests to control for any confounds introduced by the experimenter's sex[84]. All behavioral tests were performed at the end of the light cycle, between 4:00 PM and 7:00 PM. The videos were analyzed in ToxTrac[85] to track the instantaneous animal center position and quantify distance traveled. During habituation, female rats showed higher locomotion than males (two-sided unpaired $t$-test, $P = 0.0006$), therefore we normalized the distance traveled to the habituation baseline to isolate the effect of ketamine.

## [³H]DAMGO autoradiography

Rats were pretreated with naltrexone (10 mg/kg, s.c.) or vehicle 10 min prior to treatment with ketamine (10 mg/kg, i.p.) or vehicle for four consecutive days. Twenty-four hours after the last treatment, rats were

euthanized, the brains were flash frozen, and stored at -80 °C until they could be sectioned (20 μm) on a cryostat (Leica) and thaw mounted on ethanol cleaned glass slides. Slides were pre-incubated in 50 mM Tris-HCl buffer for 10 min at room temperature. The pre-incubation buffer was removed and the slides were placed in incubation buffer containing 5 nM [$^3$H]DAMGO (46 Ci/mmol, NIDA Drug Supply) for 45 min at room temperature (total binding). For non-specific binding non-tritiated DAMGO (10 μM) was also added. The sections were then washed by two 30-s washes in the Tris buffer. Finally, slides were dipped in ice cold distilled water to remove salts. After exposure to the radioligand and washing, slides were allowed to dry and were then placed into a Hypercassette™ covered by a BAS-TR2025 phosphor screen (FujiFilm; Cytiva). The slides were exposed to the phosphor screen for 12 days and then imaged using a phosphor imager (Typhoon FLA 7000; GE Healthcare). The digitized images were calibrated using $^{14}$C standard slides (American Radiolabeled Chemicals). ROIs were hand-drawn based on anatomical landmarks and radioactivity was quantified using ImageJ (NIH). The activity in 4 different sections was averaged per animal and brain region.

### General statistical analysis

Rats were randomly assigned to treatment conditions. When within-subjects factors were present in the ANOVA, Mauchly's test for sphericity was performed to determine whether the sphericity assumption was satisfied. In cases where the assumption was violated, we used a Greenhouse-Geisser adjustment to the degrees of freedom. Pairwise post-hoc comparisons were performed in case of significant ANOVA effects. In the pairwise tests, multiple comparisons were controlled using Benjamini–Hochberg false-discovery-rate (FDR) correction ($\alpha = 0.05$). All comparisons were two-tailed. We calculated effect sizes using Hedge's $g$. Statistical tests, sample sizes $n$, corrected $P$ values, and effect sizes $g$ are reported for each analysis in the text and figure captions. All statistical analyses were performed using custom scripts in R Studio and MATLAB.

### Reporting summary

Further information on research design is available in the Nature Portfolio Reporting Summary linked to this article.

## Data availability

The data generated in this study and used in all the statistical analyses are provided in the Source Data file. The raw functional ultrasound imaging data were not deposited due to the large size of each data acquisition. Access can be obtained upon request from the corresponding authors. Source data are provided with this paper.

## Code availability

Codes for all the statistical analyses are available at https://github.com/Airan-Lab/diianni2023-ketamine-fUSI/.

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

## Acknowledgements

This work was supported by a Seed Grant from the Stanford Wu Tsai Neurosciences Institute (to RDA), the NIH BRAIN Initiative (NIH/NIMH RF1MH114252 to RDA), the NIH HEAL Initiative (NIH/NINDS UG3NS115637 to RDA), the NIDA Intramural Research Program (ZIA000069 to MM), a Stanford University School of Medicine Dean's Postdoctoral Fellowship (to TDI), a Ford Foundation Fellowship Program of the National Academies of Sciences, Engineering, and Medicine (to MMA), and a National Science Foundation Graduate Research Fellowship Program (to SNE). We would like to thank Profs. Alan Schatzberg, Nolan Williams, Boris Heifets, Carolyn Rodriguez, and Robert Malenka for invaluable discussions on ketamine's effects; Profs. Jeremy Dahl, Kim Butts Pauly, and Katherine Ferrara for helpful advice on ultrasound methodologies; Dr. Keith Murphy for feedback on figure crafting; and all members of the Airan and Michaelides labs for helpful discussions.

## Author contributions

Conceptualized the study, designed and performed experiments, analyzed data, and wrote the manuscript: T.D.I. Designed and performed experiments, analyzed data: S.N.E., M.R.L., M.M.A. Performed experiments: R.C.B. Contributed to the design of experiments: M.M. Designed experiments, contributed to writing the manuscript, funding acquisition, supervision: R.D.A. All coauthors participated in the review of the manuscript and approved the final version.

## Competing interests

R.D.A. has equity and has received equity/stock options and consulting fees from Cordance Medical and Lumos Labs and grant funding from AbbVie Inc. M.M. has received research funding from AstraZeneca, Redpin Therapeutics, and Attune Neurosciences, Inc. T.D.I. has equity/stock options and received consulting fees from Attune Neurosciences, Inc. All other authors declare no conflicts of interest.
