## [Peer Review File · Nature Communications]

Sex dependence of opioid-mediated responses to subanesthetic ketamine in ratsREVIEWER COMMENTS

Reviewer #1 (Remarks to the Author):

We have reviewed a previous version of the manuscript. As discussed before, the use of functional ultrasound imaging to investigate multi-regional responses to subanesthetic ketamine is novel. The authors now added new experiments shown in Figure 2, which is an important addition to show the potential neural activity basis underlying the fUSI signal. The results will be of interest to researchers interested in ketamine's effect on brain circuitry. There remain some discussion points and statistical analyses that would strengthen the manuscript.

Comments:

- fUSI measures CBV signals across numerous brain regions. The results largely show similar regions as those covered by whole-brain cFos mapping after administration of subanesthetic ketamine (Davoudian et al., ACS Chem Neurosci 2023). The similarities and differences may be worth discussing.
- Figure 1b suggests there may be an equivalent of Fig. 1d and 1e but for the regions at bregma -3.5 mm, which would be useful to plot either in Figure 1 or in supplementary materials, as there is increasing interest in the role of RSG/RSD for ketamine's action (Vesuna et al., Nature, 2020).
- For the main results as shown in Figure 3b: if the interpretation is that opioid receptors contribute to the fUSI signals, which is delineated by the increase in fUSI signal in VEH+KET relative to NTX-KET, and that's because naltrexone blocks opioid receptor so this portion of the signal is attributed to ketamine's action on the opioid receptors. Then how does one explain for females, the increase in NTX+KET relative to VEH+KET? Here the NTX serves to block the opioid receptors, but why does NTX leads an increase from VEH+KET in females? Is there a plausible mechanism of action - as NTX should be blocking the receptors but is not causing any change? The authors should add to the discussion.
- Figure 5c and 5d - to test for sex-dependence and interactive effect of NTX and ketamine, this should be ran as a three-way ANOVA with factors of sex, drug (ketamine or vehicle) and NTX (NTX or none) and look for significance in the interactions. Currently it is tested as individual t-tests, which is known to be inaccurate - see "Erroneous analyses of interactions in neuroscience" (Nieuwenhuis et al., Nat Neurosci, 2011). The authors cannot conclude from the current data in Figure 5 that there is a sex dependence on postsynaptic density expression.

Reviewer #2 (Remarks to the Author):

In this revised version, the authors have included an important electrophysiological experiment that clarifies my main concern about the interpretation of the data.

Using CBV as a proxy for neuronal activity is possible under normal physiological conditions. However, in the context of a drug injection, it is not clear whether the increase in CBV is due to neuronal activity or a pure vascular effect induced by the drug but not correlated with neuronal activity. The fact that there are large CBV signals of 100% is also a potential problem. Indeed, it is not the first time that such high increases have been measured, but in several cases they have been associated with pathological conditions such as ictal events or cortical depolarisations.

The experiment in Supplementary Figure 3 confirms that the CBV increase is highly correlated with electrical activity, particularly in the gamma band, in agreement with previous fUS electrophysiology experiments. It's exciting to see that functional ultrasound can provide useful neuronal information

under such specific conditions.

Reviewer #3 (Remarks to the Author):

In this study, the authors administer naltrexone prior to ketamine and present fUSI data suggesting that the antidepressant-like actions of ketamine are mediated by opioid receptors in a sex-dependent manner. The authors attempted to address my earlier questions without providing further data or mechanistic analysis. Unfortunately, several of their answers rely on speculation or a small number of earlier studies, that in face value need to be replicated to establish causality within the context of the current work.

Here are but a few examples of remaining concerns. While I appreciate this is a neuroimaging study, the similar peak dCBV/CBV values for the different doses of ketamine (1, 5, 10 mg/kg) seems surprising. Since ketamine can produce many effects, are the observed findings relevant to its antidepressant action? Ketamine has a strong dose dependency and 1 mg/kg ketamine typically has not been shown to produce antidepressant-like effects. Another unaddressed concern, the focus on ketamine and MK801 produce differing effects is not entirely clear as MK801+vehicle produces a significant decrease in the peak dCBV/CBV, so MK801+veh+NTX may not be able to further reduce it given there is a floor effect, which is not addressed. Another unaddressed concern is how ketamine presumably impacts opioid receptors as a target of antidepressant action in a sex dependent manner. If ketamine is indeed binding to the mu opioid receptors to exert changes in fUSI, then the authors need to dissect this more than simply basing it on naltrexone blocking ketamine's effects, especially given the ambiguity with the MK801 findings. In the absence of such data, the study is rather correlative.

Response to the Reviewers
“Sex dependence of opioid-mediated responses to subanesthetic ketamine”
Nature Communications NCOMMS-23-17652-T

We sincerely thank the Editor and Reviewers for their careful evaluation of our work and the insightful comments.

In this document, we respond to each point raised by the Reviewers in the most recent round of review and outline the changes we have made to the text, figures, and supplementary information. We use blue font for our responses and report major changes to the text as indented paragraphs. All the changes in the attached manuscript and supplementary information are colored in red.

Reviewer 1

We have reviewed a previous version of the manuscript. As discussed before, the use of functional ultrasound imaging to investigate multi-regional responses to subanesthetic ketamine is novel. The authors now added new experiments shown in Figure 2, which is an important addition to show the potential neural activity basis underlying the fUSI signal. The results will be of interest to researchers interested in ketamine's effect on brain circuitry. There remain some discussion points and statistical analyses that would strengthen the manuscript.

We thank the Reviewer for the enthusiasm and encouraging comments. We have thoroughly revised our manuscript and have addressed all the points raised by the Reviewers. We hope that the additional experiments included in this new version will strengthen the relevance of our findings.

Comments:

1. fUSI measures CBV signals across numerous brain regions. the results largely show similar regions as those covered by whole-brain cFos mapping after administration of subanesthetic ketamine (Davoudian et al., ACS Chem Neurosci 2023). The similarities and differences may be worth discussing.

We thank the Reviewer for this recommendation. We have added a sentence to the Discussion (line 302) linking our results to the findings of Davoudian et al., ACS Chem Neurosci, 2023:

Many of these regions have been previously reported to show increased early gene (cFos) expression upon ketamine and psilocybin administration⁵⁶.

2. Figure 1b suggests there may be an equivalent of Fig. 1d and 1e but for the regions at bregma -3.5 mm, which would be useful to plot either in Figure 1 or in supplementary materials, as there is increasing interest in the role of RSG/RSD for ketamine's action (Vesuna et al., Nature, 2020).

We performed the dose-response experiments in the slice at bregma +2.5 mm to determine whether functional ultrasound imaging could resolve different doses of ketamine. To limit the number of animals used, we did not image the slice at bregma -3.5 mm in the dose response experiments.

3. For the main results as shown in Figure 3b: if the interpretation is that opioid receptors contribute to the fUSI signals, which is delineated by the increase in fUSI signal in VEH+KET relative to NTX-KET, and that's because naltrexone blocks opioid receptor so this portion of the signal is attributed to ketamine's action on the opioid receptors. Then how does one explain for females, the increase in NTX+KET relative to VEH+KET? Here the NTX serves to block the opioid receptors, but why does NTX leads an increase from VEH+KET in females? Is there a plausible mechanism of action - as NTX should be blocking the receptors but is not causing any change? The authors should add to the discussion.

In females, we did not observe any statistically significant differences in the VEH+KET vs NTX+KET comparisons. Only in the Cg1 time series we found a slight increase in the NTX+KET group compared to the VEH+KET group; however, this difference was not significant. This is also evident in the functional maps of Fig. 3c, where only two minor significant clusters were present in the anterior slice and no significant pixels were found in the posterior slice. The intra-individual peak CBV changes in Fig. 4a and the binned time series in Supplementary Figs. 4b and 5b also show no statistically significant differences between the VEH+KET and NTX+KET groups in females.

4. Figure 5c and 5d - to test for sex-dependence and interactive effect of NTX and ketamine, this should be ran as a three-way ANOVA with factors of sex, drug (ketamine or vehicle) and NTX (NTX or none) and look for significance in the interactions. Currently it is tested as individual t-tests, which is known to be inaccurate - see "Erroneous analyses of interactions in neuroscience" (Nieuwenhuis et al., Nat Neurosci, 2011). The authors cannot conclude from the current data in Figure 5 that there is a sex dependence on postsynaptic density expression.

We thank the Reviewer for this comment. In the previous version of the manuscript, we performed a two-way ANOVA with factors of treatment and sex, and we reported the results in the main text but not in the Figure caption. To further confirm our previous conclusions, we are now including a three-way ANOVA with factors of sex, pretreatment (NTX or VEH), and treatment (KET or VEH). We confirm that in the PSD-95 data we observe significant effects of sex ($P = 6.06e-06$), pretreatment ($P = 3.62e-04$), treatment ($P = 5.63e-07$), and interactions including sex \times pretreatment \times treatment ($P = 5.55e-04$). Conversely, we do not observe any statistically significant effect in the DAPI data.

We added the following paragraph in the main text at line 211:

In the quantified PSD-95, a three-way ANOVA with factors of sex, pretreatment (NTX or VEH), and treatment (KET or VEH) revealed significant effects of sex ($F_{1,24} = 33.25$, $P = 6.06E-06$), pretreatment ($F_{1,24} = 17.21$, $P = 3.62E-04$), and treatment ($F_{1,24} = 45.49$, $P = 5.63E-07$), and significant sex \times treatment ($F_{1,24} = 17.21$, $P = 3.62E-04$), sex \times pretreatment ($F_{1,24} = 16.38$, $P = 4.68E-04$), treatment \times pretreatment ($F_{1,24} = 42.36$, $P = 9.89E-07$), and sex \times pretreatment \times treatment ($F_{1,24} = 15.84$, $P = 5.55E-04$) interactions. Subanesthetic ketamine increased the expression of PSD-95 in the mPFC of both male (two-tailed unpaired t -test; corrected $P = 1.42E-05$) and female ($P = 0.0071$) rats compared to the saline-injected controls (**Figure 5c-d**). In agreement with our imaging findings, naltrexone pretreatment completely blocked ketamine's effect in male rats ($P = 1.86E-05$). Meanwhile, neither males nor females receiving naltrexone alone nor naltrexone and ketamine showed significant differences compared to the saline-injected controls.

Importantly, there were no significant differences in the nuclear DAPI staining, considered as a control.

Reviewer 2

In this revised version, the authors have included an important electrophysiological experiment that clarifies my main concern about the interpretation of the data.

Using CBV as a proxy for neuronal activity is possible under normal physiological conditions. However, in the context of a drug injection, it is not clear whether the increase in CBV is due to neuronal activity or a pure vascular effect induced by the drug but not correlated with neuronal activity. The fact that there are large CBV signals of 100% is also a potential problem. Indeed, it is not the first time that such high increases have been measured, but in several cases they have been associated with pathological conditions such as ictal events or cortical depolarisations.

The experiment in Supplementary Figure 3 confirms that the CBV increase is highly correlated with electrical activity, particularly in the gamma band, in agreement with previous fUS electrophysiology experiments. It's exciting to see that functional ultrasound can provide useful neuronal information under such specific conditions.

We thank the Reviewer for the enthusiasm and for highlighting the potential of our pharmaco-fUSI investigations to investigate circuit-level neural mechanisms of subanesthetic ketamine.

Reviewer 3

In this study, the authors administer naltrexone prior to ketamine and present fUSI data suggesting that the antidepressant-like actions of ketamine are mediated by opioid receptors in a sex-dependent manner. The authors attempted to address my earlier questions without providing further data or mechanistic analysis. Unfortunately, several of their answers rely on speculation or a small number of earlier studies, that in face value need to be replicated to establish causality within the context of the current work.

Comments:

1. While I appreciate this is a neuroimaging study, the similar peak dCBV/CBV values for the different doses of ketamine (1, 5, 10 mg/kg) seems surprising. Since ketamine can produce many effects, are the observed findings relevant to its antidepressant action? Ketamine has a strong dose dependency and 1 mg/kg ketamine typically has not been shown to produce antidepressant-like effects.

We thank the reviewer for this comment. We performed the dose response study with 0, 1, 5, and 10 mg/kg ketamine to assess if fUSI was able to resolve increasing doses of ketamine. To this end, we found a strong dose-dependent relationship both in the peak CBV measure in Figure 1e (two-way mixed-effects ANOVA; within-subjects factor of region, $F_{3,08,92.55} = 17.82$, $P = 2.33E-09$; between-subjects factor of dose, $F_{3,30} = 8.26$, $P = 3.74E-04$; interaction, $F_{9,25,92.55} = 3.38$, $P = 0.001$) and in the area under the curve in Supplementary Fig. 2b (two-way mixed-effects ANOVA; within-subjects factor of region, $F_{2,78,83.35} = 5.32$, $P = 0.003$; between-subjects factor of dose, $F_{3,30} = 10.47$, $P = 7.11E-05$; interaction, $F_{8,33,83.35} = 2.62$, $P = 0.012$). Therefore, we agree with the Reviewer's statement that "ketamine has a strong dose dependency", as clearly suggested by our fUSI data. Once we established that we could resolve different doses of ketamine with the fUSI imaging modality, we proceeded to perform all our investigations with a 10 mg/kg ketamine

dose, since this dose produces antidepressant-like effects in rat behavioral models in previously published literature.

We also point out that, like the fUSI measures, our electrocortigraphy recordings also show comparable peak values with the 1 and 10 mg/kg ketamine doses, in particular in the gamma frequency band, although differing areas under the curves in both measures between these doses.

2. Another unaddressed concern, the focus on ketamine and MK801 produce differing effects is not entirely clear as MK801+vehicle produces a significant decrease in the peak dCBV/CBV, so MK801+veh+NTX may not be able to further reduce it given there is a floor effect, which is not addressed.

To address the concern of a possible floor effect potentially limiting the ability of fUSI to resolve dCBV/CBV signal attenuations induced by naltrexone in the MK-801 experiment, we have repeated the fUSI experiments with a higher dose of MK-801 (0.25 mg/kg). Confirming our previous findings with 0.1 mg/kg, this higher dose also did not show any statistically significant effects of naltrexone pretreatment, as we note in the main text at lines 194-195 and in the caption of Supplementary Fig. 8:

Two-way ANOVA; within-subjects factor of region, $F_{9,54} = 2.08$, $P = 0.048$; within-subjects factor of treatment, $F_{1,6} = 0.02$, $P = 0.9$; interaction, $F_{9,54} = 0.56$, $P = 0.8$. Two-sided paired t -tests (VEH+MK-801 vs NTX+MK-801) showed no significant effects.

Therefore, we believe a floor effect is unlikely. We also point out that in our prior study (Levinstein et al., *Biological Psychiatry*, **93**, 1118–1126, 2023), we showed a significant reduction by naltrexone pretreatment in the dCBV/CBV signal evoked in the nucleus accumbens by a single reinforcing dose of 0.5 mg/kg S-ketamine, even though the original dCBV/CBV signal was <40%. Below we include the relevant figure from this paper for reference.

Figure: Functional ultrasound imaging figure from Levinstein et al., *Biological Psychiatry*, **93**, 1118–1126, 2023, showing that fUSI could resolve CBV signal reductions produced by naltrexone pretreatment even though the original CBV signal evoked by an S-ketamine dose of 0.5 mg/kg was less than 40%.

3. Another unaddressed concern is how ketamine presumably impacts opioid receptors as a target of antidepressant action in a sex dependent manner. If ketamine is indeed binding to the mu opioid receptors to exert changes in fUSI, then the authors need to dissect this more than simply basing it on naltrexone blocking ketamine's effects, especially given the ambiguity with the MK801 findings. In the absence of such data, the study is rather correlative.

We thank the reviewer for this comment, as it gave us an opportunity to perform additional experiments in an effort to elucidate the mechanisms of ketamine and naltrexone binding at the opioid receptors level, whose results raise an important consideration that the sex-differences we observe may be more due to differences in the response of the opioid system between male and female rats. Specifically, we performed in vitro autoradiography with the selective mu opioid receptor agonist [³H]DAMGO in male and female rats receiving repeated naltrexone and ketamine infusions. The results of this autoradiography experiment are reported in Figure 7 in the main manuscript.

We added the following paragraphs starting at line 249:

Chronic naltrexone upregulates mu opioid receptors in female rats

We next investigated the molecular mechanisms underlying the opioid-dependent behavioral adaptations induced by chronic dosing of ketamine and naltrexone in male rats. To this end, we performed in vitro autoradiography with the selective mu opioid receptor (MOR) agonist [³H]DAMGO, following our previous study showing significant modulations of MOR density in key brain regions specifically with the (S)-ketamine isomer¹⁷. Male and female rats were pretreated with naltrexone (10 mg/kg, s.c.) or vehicle 10 minutes prior to ketamine (10 mg/kg, i.p.) or vehicle for four consecutive days to mirror our behaviorally relevant dosing protocol (**Figure 6**). Twenty-four hours after the last injection the animals were euthanized, and brain slices were incubated with [³H]DAMGO and imaged using a phosphor imager. We have previously shown that repeated (S)-ketamine infusions (20 mg/kg/day, iv for 8 days) decreased MOR density in the mPFC, NAc, and thalamus in male and female rats¹⁷. Curiously, here we did not observe a reduction in MOR density in either males or females following daily injections of 10 mg/kg racemic ketamine (two-tailed unpaired *t*-test, corrected $P > 0.217$) (**Figure 7**). However, we found a statistically significant effect of sex in the NAc (two-way ANOVA; $F_{1,40} = 5.12$, $P = 0.029$) (**Figure 7b**), and post-hoc analysis in this region revealed statistically significant differences with naltrexone treatment only in female rats (both in the NTX+KET vs VEH+KET and NTX+VEH vs VEH+KET comparisons; two-tailed unpaired *t*-test, corrected $P < 0.036$). In addition, there were statistically significant differences between males and females in the NTX+VEH group in the NAc (two-tailed unpaired *t*-test, corrected $P < 0.039$), and in the NTX+KET group in the CPu (two-tailed unpaired *t*-test, corrected $P < 0.027$). Our results suggest that chronic naltrexone administration increased MOR density in the NAc selectively in female rats.

This effect could potentially explain why ketamine-induced locomotor sensitization was not blocked by naltrexone pretreatment in females: while naltrexone fully blocked MORs in male rats preventing behavioral sensitization, females had increased opioid system capacitance to increase MOR density in response to chronic naltrexone exposure, and as a result, racemic ketamine could still actuate a sufficient number of receptors in female rats to elicit a sensitized locomotor effect. Importantly, although we did not find significantly decreased MOR density from 10 mg/kg racemic ketamine (equivalent to 5 mg/kg (S)-ketamine) administration compared to vehicle, we observed a mean reduction of 9.8% in [³H]DAMGO binding in the NAc when we aggregated male and female rats. For comparison, in our previous study the binding potential in the NAc was reduced by 29.5% in male and female rats after 8 daily i.v. infusions of 20 mg/kg (S)-ketamine¹⁷. While it is difficult to directly compare these results due to the different dosing regimen, it is worth noting that in our prior study the cumulative dose of (S)-ketamine was 8x higher (160 mg/kg vs 20 mg/kg assuming 50% (S)-ketamine isomer in the racemic formulation).

Therefore, considering that (S)- but not (R)-ketamine binds at the MOR site⁴⁵, it was expected that we would observe a lesser reduction in MOR density in this current experiment.

We also added the following paragraph to the Discussion:

Finally, we showed that the lack of ketamine-evoked behavioral adaptations in females may be explained by a compensatory effect of repeated naltrexone infusions, which increased MOR density in the NAc in female rats only. Previous studies reported upregulation of MORs following chronic administration of naltrexone or naloxone^{66,67}. Here we add to this knowledge by highlighting a clear sex dependence of these effects. Synthesizing our current results with those in the literature, including the differential pharmacokinetics of ketamine in male and female humans and rodents {Highland et al, current ref 69}, suggests that at doses that we observed to be behaviorally relevant, ketamine does indeed act partly through opioid pathways to induce varied physiologic, cellular, and behavioral-level effects, but that female rats have a greater capacitance in opioid pathways to compensate in the setting of opioid blockade, so that such opioid blockade does not appreciably modulate these varied effects of subanesthetic racemic ketamine. We posit that this differential capacitance for modulating opioid signaling between varied subject populations may account for the heterogeneity of results across clinical trials that have attempted to investigate the role of the opioid pathway in the affective effects of subanesthetic ketamine {current refs 10-13}.

Together with the strengthened MK-801 findings discussed above, we believe that these additional, independent experiments will contribute to an improved understanding of ketamine's pharmacology, expand our knowledge of its sex-dependent responses, and guide future clinical and preclinical investigations of ketamine and its opioid-mediated effects.

REVIEWERS' COMMENTS

Reviewer #1 (Remarks to the Author):

The authors have addressed my comments, and I do not have anything to add. The use of functional ultrasound imaging to study neural responses to subanesthetic ketamine is new, and this paper will provide useful information for those researchers who are interested in understanding the mechanisms behind ketamine's drug actions.

Reviewer #2 (Remarks to the Author):

Dear Editor,

The authors answered all the questions in the previous round, including some new control experiments. I recommend the mss for publication.

Reviewer #3 (Remarks to the Author):

The authors attempted to address my earlier concerns but the revision raises a number of questions, which fundamentally obfuscates the conclusions of this study. Given the confusing nature of the findings and their incongruence with ketamine's antidepressant effects seen in the clinic, I cannot recommend this study for publication.

1. The concern regarding previous comment #1 is not addressed. While ketamine has a 'strong dose dependency' as shown by the dCBV/CBV values at 1, 5, 10 mg/kg, however, antidepressant effects are not observed in rats at 1 mg/kg, making it is doubtful that these effects are relevant to ketamine's antidepressant action. As it stands, one could make the claim that the presented data is important for phenomena related to ketamine's psychotomimetic effects as clinical work has shown a strong dose-dependence to dissociative symptoms in patients (Fava and colleagues, 2020).
2. Ketamine's rapid antidepressant effects do not show significant sex differences in the clinic. Again, the authors' observations using opioid antagonists point to possible impact of ketamine action on processes other than its antidepressant action.
3. It is unclear what the experiments examining repeated ketamine administration (10 mg/kg; 4 daily sessions) with and without naltrexone on locomotor sensitization and mu opioid receptor levels is examining. The daily dosing paradigm is not similar to that used for the treatment of affective disorders.

Response to the Reviewers
“Sex dependence of opioid-mediated responses to subanesthetic ketamine in rats”
Nature Communications NCOMMS-23-17652-T

We sincerely thank the Editor and Reviewers for their careful evaluation of our work and the insightful comments.

In this document, we respond to each point raised by the Reviewers in the most recent round of review. We use **blue** font for our responses and report major changes to the text as indented paragraphs. All the changes in the attached manuscript and supplementary information are colored in **red**.

Reviewer 1

The authors have addressed my comments, and I do not have anything to add. The use of functional ultrasound imaging to study neural responses to subanesthetic ketamine is new, and this paper will provide useful information for those researchers who are interested in understanding the mechanisms behind ketamine's drug actions.

We thank the Reviewer immensely for their thoughtful reviews throughout this process and their current enthusiasm and encouragement.

Reviewer 2

Dear Editor,

The authors answered all the questions in the previous round, including some new control experiments. I recommend the mss for publication.

We thank the Reviewer for their past reviews, which have served to greatly improve our manuscript.

Reviewer 3

The authors attempted to address my earlier concerns but the revision raises a number of questions, which fundamentally obfuscates the conclusions of this study. Given the confusing nature of the findings and their incongruence with ketamine's antidepressant effects seen in the clinic, I cannot recommend this study for publication.

1. The concern regarding previous comment #1 is not addressed. While ketamine has a 'strong dose dependency' as shown by the dCBV/CBV values at 1, 5, 10 mg/kg, however, antidepressant effects are not observed in rats at 1 mg/kg, making it is doubtful that these effects are relevant to ketamine's antidepressant action. As it stands, one could make the claim that the presented data is important for phenomena related to ketamine's psychotomimetic effects as clinical work has shown a strong dose-dependence to dissociative symptoms in patients (Fava and colleagues, 2020).

As it appears to be a point of confusion, we have taken care to emphasize that following the initial dose-response characterization, we used 10 mg/kg as the ketamine dose for all the experiments in this study, as this dose has been used to influence affective behaviors in a variety of rodent studies.

New lines 133-136 now are:

Following these initial methodologic and dose-response characterizations of using fUSI to study the effects of subanesthetic ketamine, we selected 10 mg/kg as the dose of ketamine for subsequent experiments, following prior rodent studies of the affective effects of ketamine that show reliable behavioral efficacy with this dose⁹.

2. Ketamine's rapid antidepressant effects do not show significant sex differences in the clinic. Again, the authors' observations using opioid antagonists point to possible impact of ketamine action on processes other than its antidepressant action.

We concede that the influence of sex on the actions of subanesthetic ketamine may express differently in rodents and humans. However, none of the clinical studies to date that have specifically evaluated ketamine in the setting of opioid antagonists have been powered to detect sex-based differences. Given our findings we believe it to be appropriate to suggest that future clinical studies of the potential interaction of the actions of opioids and ketamine should be powered with respect to key biological variables such as sex. We have retooled the discussion section of the manuscript to more clearly highlight that this is a rodent study, with the associated caveat that these results would need to be specifically validated in clinical populations.

The relevant sections of our Discussion have been adjusted to add:

At lines 339-341:

It is important to note that these results from rats would need to be explicitly verified in clinical trials before drawing meaningful conclusions with respect to clinical treatments using subanesthetic ketamine.

At lines 363-365:

There are several limitations to our study. First, as we have noted, these results from rodents would need explicit verification in controlled clinical trials to draw meaningful conclusions for clinical treatments using ketamine.

3. It is unclear what the experiments examining repeated ketamine administration (10 mg/kg; 4 daily sessions) with and without naltrexone on locomotor sensitization and mu opioid receptor levels is examining. The daily dosing paradigm is not similar to that used for the treatment of affective disorders.

This locomotor sensitization paradigm has been used in a variety of rodent studies to assess the behavioral adaptive response to psychotropic drugs, especially those with opioidergic actions, like racemic ketamine and its isoforms have been shown to be in prior studies (e.g. Bonaventura et al. *Molecular Psychiatry*. 2021.). In addition, repeated ketamine dosing is used to maximize antidepressant response rates at many sites. For instance, Phillips et al. (*American Journal of Psychiatry*, 2019) showed that a ketamine regimen of a loading phase of thrice weekly (i.e., every other day) dosing for two weeks, followed by a maintenance phase of once weekly ketamine for four weeks was able to improve overall response rates in depressed patients. We therefore see the behavioral evaluation of repeated ketamine dosing as highly relevant given the use of

repeated ketamine dosing clinically. Regarding the higher frequency of dosing in rats, we note that rodents have shown a higher rate of ketamine metabolism compared to humans, with net faster clearance kinetics, as evidenced by the order of magnitude higher dosing for affective efficacy that is typical in rodent studies (10 mg/kg in this and other rodent studies) versus humans (typically 0.5 mg/kg).